# Kinesin-5-independent mitotic spindle assembly requires the antiparallel microtubule crosslinker Ase1 in fission yeast

Sergio A. Rincon[1], Adam Lamson[2], Robert Blackwell[2], Viktoriya Syrovatkina[3], Vincent Fraisier[1], Anne Paoletti[1], Meredith D. Betterton[2] & Phong T. Tran[1,3]

Bipolar spindle assembly requires a balance of forces where kinesin-5 produces outward pushing forces to antagonize the inward pulling forces from kinesin-14 or dynein. Accordingly, Kinesin-5 inactivation results in force imbalance leading to monopolar spindle and chromosome segregation failure. In fission yeast, force balance is restored when both kinesin-5 Cut7 and kinesin-14 Pkl1 are deleted, restoring spindle bipolarity. Here we show that the *cut7Δpkl1Δ* spindle is fully competent for chromosome segregation independently of motor activity, except for kinesin-6 Klp9, which is required for anaphase spindle elongation. We demonstrate that *cut7Δpkl1Δ* spindle bipolarity requires the microtubule antiparallel bundler PRC1/Ase1 to recruit CLASP/Cls1 to stabilize microtubules. Brownian dynamics-kinetic Monte Carlo simulations show that Ase1 and Cls1 activity are sufficient for initial bipolar spindle formation. We conclude that pushing forces generated by microtubule polymerization are sufficient to promote spindle pole separation and the assembly of bipolar spindle in the absence of molecular motors.

[1] Institut Curie, PSL Research University, CNRS, UMR 144, F-75005 Paris, France. [2] Department of Physics, University of Colorado, Boulder, Colorado 80309, USA. [3] Department of Cell & Developmental Biology, University of Pennsylvania, Philadelphia, Pennsylvania 19104, USA. Correspondence and requests for materials should be addressed to M.D.B. (email: mdb@colorado.edu) or to P.T.T. (email: phong.tran@curie.fr).

Chromosome segregation is an essential cellular process, where absolute fidelity is required, as errors may lead to developmental defects and diseases. Chromosome segregation requires the assembly of the spindle, a microtubule (MT)-based structure which efficiently captures and segregates sister chromatids during mitosis. MT minus ends converge at the spindle poles, while MT plus ends emanating from the opposite pole interdigitate at the spindle midzone. Kinesin-5 family members are plus-end directed motors organized as homotetramers that crosslink antiparallel MTs and slide them apart, pushing on the spindle poles, and therefore promoting spindle bipolarity and elongation[1–3]. In most systems the lack of kinesin-5 activity results in spindle collapse into a monopolar structure unable to segregate sister chromatids[4–7]. Minus-end directed motors such as kinesin-14 members or dynein have been proposed to counter-balance kinesin-5 pushing forces by localizing at the spindle poles, pulling on MTs emanating from the opposite poles, producing inward forces to bring the spindle poles together[8,9]. Proper co-ordination of these two activities is required for the correct assembly of the spindle and the control of its length[10]. Indeed, depletion of minus-end directed motors producing inward pulling forces allows spindle assembly in the absence of kinesin-5 activity in human cells, *Xenopus* eggs, *Drosophila* and fission yeast[11–16]. Therefore, additional mechanisms must operate to orchestrate spindle assembly in a kinesin-5-independent manner.

In this work, we show that fission yeast bipolar spindle assembly and chromosome segregation can take place in the absence of both kinesin-5 Cut7 and kinesin-14 Pkl1. In *cut7Δpkl1Δ* cells, spindle formation and bipolarity requires the MT antiparallel bundler Ase1 (refs 17,18) to recruit Cls1 (refs 19,20) to stabilize MTs. Indeed, no motors are needed for initial bipolar spindle formation. Computer simulation shows that the activity of Ase1 and Cls1 is sufficient for spindle bipolarity, suggesting that MT polymerization drives pole separation in the absence of motors.

## Results

***cut7Δpkl1Δ* cells assemble short bipolar spindles.** To verify that kinesin-5 and kinesin-14 contribute to the balance of spindle forces, we tested if the monopolar spindles produced by kinesin-14 Pkl1 overexpression were rescued by co-overexpression of kinesin-5 Cut7 (ref. 21). The results showed that increased Cut7 expression restored spindle bipolarity in Pkl1 overexpressing cells (Supplementary Fig. 1a,b), indicating that Cut7 and Pkl1 are antagonistic. To gain insight on the kinesin-5-independent mechanism of spindle assembly, we analysed spindle assembly dynamics and chromosome segregation by live-cell imaging in fission yeast double-deletion *cut7Δpkl1Δ* cells expressing mCherry-Atb2 (α-tubulin) and Sid4-green fluorescent protein (GFP; spindle pole body (SPB) component) compared to wild-type cells. Wild-type cells exhibited stereotypical three-phase spindle elongation kinetics, corresponding to prophase (phase I), metaphase (phase II), and anaphase A (chromatid separation) and B (spindle elongation) (phase III; Fig. 1a,b). *cut7Δpkl1Δ* cells assembled shorter metaphase spindles (>2-fold reduction compared to wild-type cells) that eventually elongated with a delayed transition to anaphase B by over 10 min compared to wild-type cells (Fig. 1a–d). Noteworthy, spindle elongation velocity during anaphase B was reduced in *cut7Δpkl1Δ* cells (30% reduction compared to wild-type cells; Fig. 1e), suggesting that Cut7 participates in anaphase spindle elongation in addition to its established function in spindle bipolarity.

To check if the spindle assembled in the *cut7Δpkl1Δ* cells is competent for chromosome segregation, we used the kinetochore marker Mis12-GFP and found that sister kinetochores moved poleward concomitantly with spindle elongation, similar to wild-type cells (Fig. 1g). We also checked the spindle level of cyclin B Cdc13, which drops on chromosome bi-orientation[22]. Cdc13 disappeared from *cut7Δpkl1Δ* spindles right before spindle elongation (Fig. 1h). This suggests that the delayed transition to anaphase in *cut7Δpkl1Δ* cells is due to chromosome bi-orientation defects. In agreement with these results, deletion of the spindle assembly checkpoint (SAC) component Mad2 in *cut7Δpkl1Δ* cells resulted in colonies growing very poorly (Fig. 1f), indicating that *cut7Δpkl1Δ* cells require the additional time provided by the SAC to properly capture and segregate chromosomes. These data indicate that if the SAC is active, fission yeast cells manage to assemble a functional mitotic spindle able to properly segregate chromosomes when the counteracting kinesins Cut7 and Pkl1 are deleted.

**Klp9 is not involved in spindle assembly in *cut7Δpkl1Δ* cells.** We next wondered which proteins may be responsible for spindle assembly in *cut7Δpkl1Δ* cells. The kinesin-6 Klp9, the major outward pushing force motor that slides antiparallel MTs apart during anaphase spindle elongation was a good candidate[23]. Tetrad dissection indeed showed that *klp9* is essential in *cut7Δpkl1Δ* cells (Fig. 2a). Nevertheless, time-lapse movies revealed that Klp9 was recruited to *cut7Δpkl1Δ* spindles just before their elongation in anaphase B (Fig. 2b), after Cdc13 degradation (Supplementary Fig. 2a), suggesting that Klp9 does not participate in the initial phases of spindle assembly and bipolarity establishment.

We further analysed spindle behaviour in a *klp9* shut-off strain (*cut7Δpkl1Δklp9^On/Off^*). On Klp9 shut-off, these cells assembled bipolar spindles in prophase which failed to efficiently elongate during anaphase (Fig. 2c,d), resulting in chromosome 'cut' by the cytokinetic machinery in 100% of the cells (data not shown). Importantly, metaphase spindle length and the transition time to anaphase B were similar in *cut7Δpkl1Δklp9^Off^* and *cut7Δpkl1Δ* cells (Fig. 2e,f), demonstrating that Klp9 does not participate in spindle assembly in the absence of Cut7.

We next checked whether *klp9* overexpression (*klp9^OE^*) could favour spindle assembly in *cut7Δpkl1Δ* cells. We found that *klp9^OE^* does not increase the metaphase spindle length nor shorten the transition time to anaphase (Fig. 2e,f). These data are also consistent with the fact that *klp9^OE^* does not rescue the *cut7^24^* mutant (Supplementary Fig. 2b). Nevertheless, *klp9^OE^* restored the velocity of anaphase spindle elongation to wild-type values (Fig. 2g,h). Altogether, these results show that, although Cut7 and Klp9 are both capable of sliding antiparallel MTs apart, only Cut7 functions in bipolar spindle assembly before the anaphase transition.

To determine if the lack of anaphase spindle elongation in *cut7Δpkl1Δklp9^On/Off^* cells upon Klp9 shut-off was due to the absence of Cut7 or Pkl1, we analysed anaphase spindle elongation velocity in wild-type, *pkl1Δ*, *klp9Δ* and *pkl1Δklp9Δ* cells. We could not detect significant differences in anaphase spindle elongation between the wild-type and *pkl1Δ* cells nor *klp9Δ* and *pkl1Δklp9Δ* cells (Supplementary Fig. 2d–f). Therefore, we conclude that Klp9 and Cut7 are the major contributors to spindle elongation during anaphase. This may explain why the deletion of *klp9* is synthetic sick with *cut7^24^* mutant (Supplementary Fig. 2b).

Since Klp9 is not responsible for spindle bipolarity in *cut7Δpkl1Δ* cells, we wondered whether another mitotic kinesin might contribute to spindle assembly. To answer this question, we created a strain lacking all mitotic kinesins (Cut7, Pkl1, Klp2, Klp6 and Klp5)—except Klp9, which is essential for anaphase spindle elongation. Cells lacking all mitotic kinesins assembled

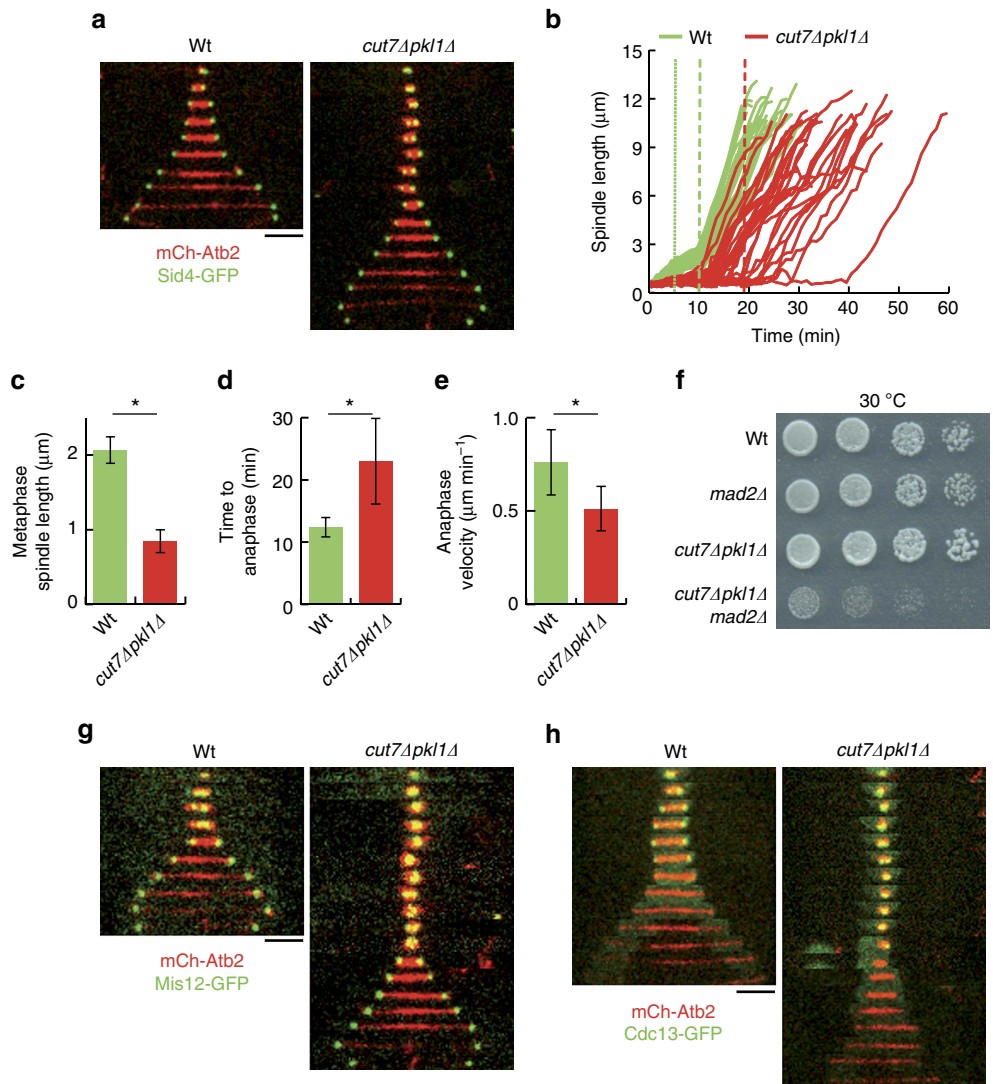

**Figure 1 | cut7Δpkl1Δ cells assemble a functional spindle.** (**a**) Time-lapse images of wild-type (wt) and cut7Δpkl1Δ cells expressing mCherry-Atb2 (tubulin) and Sid4-GFP (spindle pole marker) from mitosis onset to spindle breakdown. Each frame corresponds to 3 min interval. (**b**) Comparative plot of spindle length dynamics of wild-type (green curves; n = 19) and cut7Δpkl1Δ cells (red curves; n = 24). Dotted vertical line corresponds to phase I–II transition; dashed vertical line corresponds to phase II–III transition. (**c**) Bar plot showing metaphase spindle length at anaphase transition, reported by Cdc13-GFP degradation, of wild-type (green; n = 15) and cut7Δpkl1Δ cells (red; n = 24). *P < 0.001. (**d**) Bar plot showing the time to anaphase transition in wild-type (green; n = 15) and cut7Δpkl1Δ cells (red; n = 24). *P < 0.001. (**e**) Bar plot showing anaphase spindle elongation velocity of wild-type (green; n = 29) and cut7Δpkl1Δ cells (red; n = 22). *P < 0.001. (**f**) Serial dilution (fourfold) assay showing the genetic interaction of mad2Δ with the cut7Δpkl1Δ strain. Plates were incubated 2–3 days at the specified temperatures. (**g**) Time-lapse images of wild-type (wt) and cut7Δpkl1Δ cells expressing mCherry-Atb2 and Mis12-GFP (kinetochore marker) from mitosis onset to spindle breakdown. Each frame corresponds to 3 min interval. (**h**) Time-lapse images of wild-type (wt) and cut7Δpkl1Δ cells expressing mCherry-Atb2 and Cdc13-GFP (cyclin B) from mitosis onset to spindle breakdown. Each frame corresponds to 3 min interval. Data show mean ± s.d. and Student's t-test P value. Scale bars, 2 μm.

spindles that displayed similar dynamics to that cut7Δpkl1Δ spindles (Fig. 2i), albeit with more pronounced MT protrusions (Supplementary Fig. 2c), likely due to the absence of klp5-klp6 MT depolymerization activity[24,25]. These data show that the mitotic spindle can be assembled in the absence of all mitotic kinesins in fission yeast. This conclusion was unexpected. Therefore, we chose to further explore mechanisms of motor-independent spindle assembly through a combination of in vivo experiments and in silico simulations.

**Ase1 and Cls1 are critical for spindle assembly in cut7Δpkl1Δ cells.** Spindle assembly requires the crosslinking of MTs emanating from the opposite poles. We reasoned that other MT crosslinkers

might become essential in the absence of Cut7. Ase1 is a non-essential antiparallel MT bundler, whose deletion results in metaphase spindle instability[17,18,26,27] (Supplementary Fig. 3a), short spindle length at anaphase transition[28] (Supplementary Fig. 3b) and erratic transition time to anaphase (Supplementary Fig. 3c). Consistent with a role in metaphase, Ase1 was recruited to the spindle before anaphase both in the wild-type and cut7Δpkl1Δ cells (Supplementary Fig. 3d). Tetrad dissection further revealed that ase1 is essential in cut7Δpkl1Δ cells (Fig. 3a). To determine the role of Ase1 in spindle assembly, we created an ase1 shut-off strain (cut7Δpkl1Δase1^On/Off) expressing the nucleoporin Cut11-GFP to visualize the nuclear envelope (NE) and analysed spindle behaviour upon ase1 shut-off (Fig. 3b). Interestingly, ∼40% of cut7Δpkl1Δase1^Off cells failed to fully

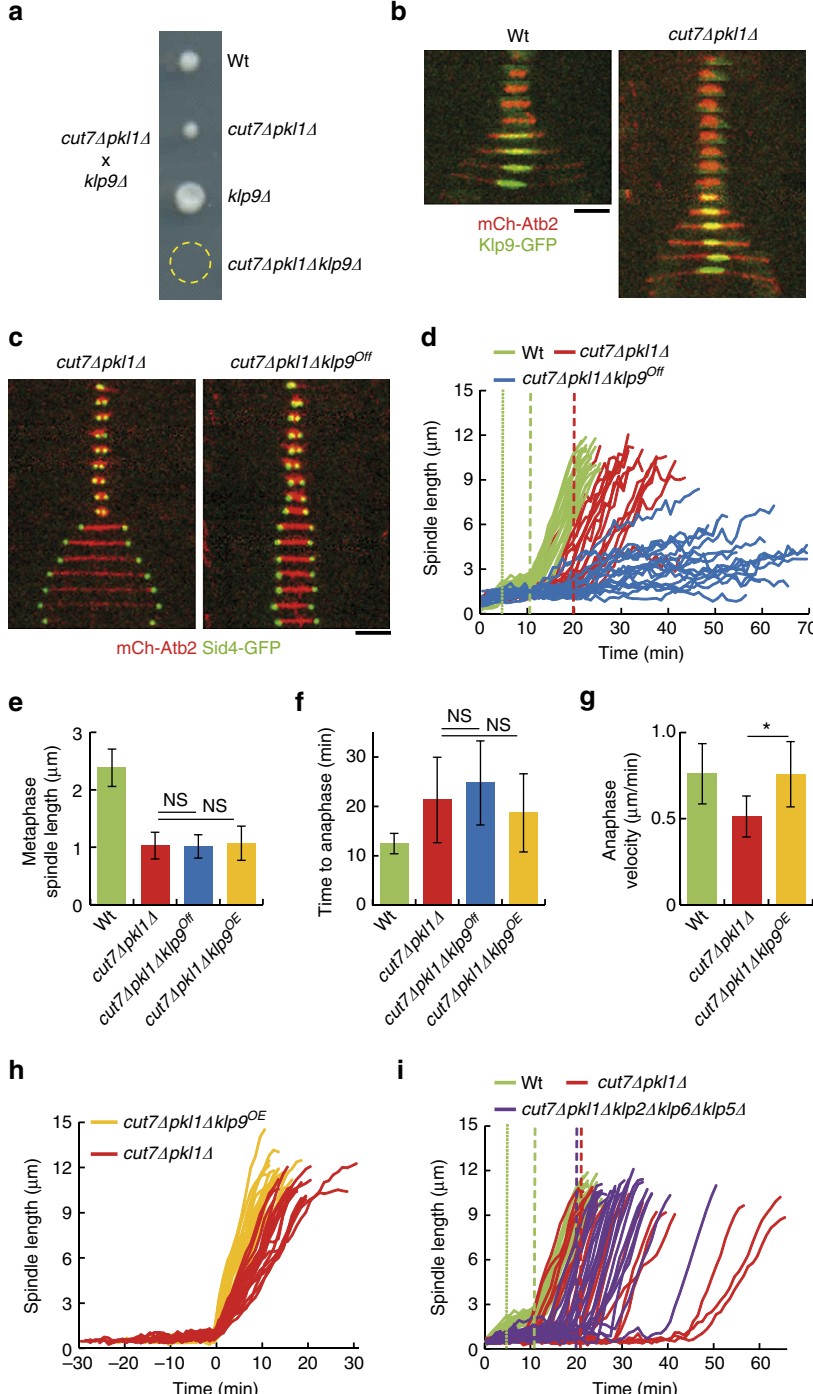

**Figure 2 | Klp9 is not essential for spindle assembly in *cut7Δpkl1Δ* cells.** (**a**) Tetrad dissection of *cut7Δpkl1Δ* cells crossed to *klp9Δ* cells. (**b**) Time-lapse images of wild-type (wt) and *cut7Δpkl1Δ* cells expressing mCherry-Atb2 and Klp9-GFP from mitosis onset to spindle breakdown. Each frame corresponds to 3 min interval. (**c**) Time-lapse images of *cut7Δpkl1Δ* and *cut7Δpkl1Δklp9^Off* cells expressing mCherry-Atb2 and Sid4-GFP from mitosis onset to spindle breakdown, after 8 h of repression/shut-off. Each frame corresponds to 3 min interval. (**d**) Comparative plot of spindle length dynamics *cut7Δpkl1Δ* (red curves; n = 16) and *cut7Δpkl1Δklp9^Off* cells (blue curves; n = 18) after 8 h of repression/shut-off. Dotted vertical line corresponds to phase I–II transition; dashed vertical line corresponds to phase II–III transition. (**e**) Bar plot showing metaphase spindle length at anaphase transition, reported by Cdc13-GFP degradation, of wild-type (green; n = 43), *cut7Δpkl1Δ* (red; n = 36), *cut7Δpkl1Δklp9^Off* cells after 8 h repression/shut-off (blue; n = 26), and *cut7Δpkl1Δklp9^OE* cells after 20 h of overexpression induction (orange; n = 37). ns, not statistically significant. (**f**) Bar plot showing anaphase transition time of wild-type (green; n = 43), *cut7Δpkl1Δ* (red; n = 36), *cut7Δpkl1Δklp9^Off* cells after 8 h incubation in thiamine-containing medium (blue; n = 26) and *cut7Δpkl1Δklp9^OE* cells after 20 h of overexpression induction (orange; n = 37). NS, not statistically significant. (**g**) Bar plot showing anaphase spindle elongation velocity of wild-type (green; n = 29), *cut7Δpkl1Δ* cells (red; n = 22) and *cut7Δpkl1Δklp9^OE* cells after 20 h of overexpression induction (orange; n = 29). *P < 0.001. (**h**) Comparative plot of spindle length dynamics of *cut7Δpkl1Δ* (red curves; n = 20) and *cut7Δpkl1Δklp9^OE* cells (orange curves; n = 20) after 20 h of overexpression induction. Time 0 represents the start of anaphase elongation. (**i**) Comparative plots of spindle length dynamics of *cut7Δpkl1Δ* (red curves; n = 22) and *cut7Δpkl1Δklp2Δklp6Δklp5Δ* cells (purple curves; n = 22). Dotted vertical line corresponds to phase I–II transition; dashed vertical line corresponds to phase II–III transition. Data show mean ± s.d. and Student's *t*-test *P* value. Scale bars, 2 μm.

elongate the spindle in anaphase, while another 40% produced monopolar spindles, resulting in a 'cut' phenotype in both cases (Fig. 3b,c). The remaining 20% successfully elongated the spindle

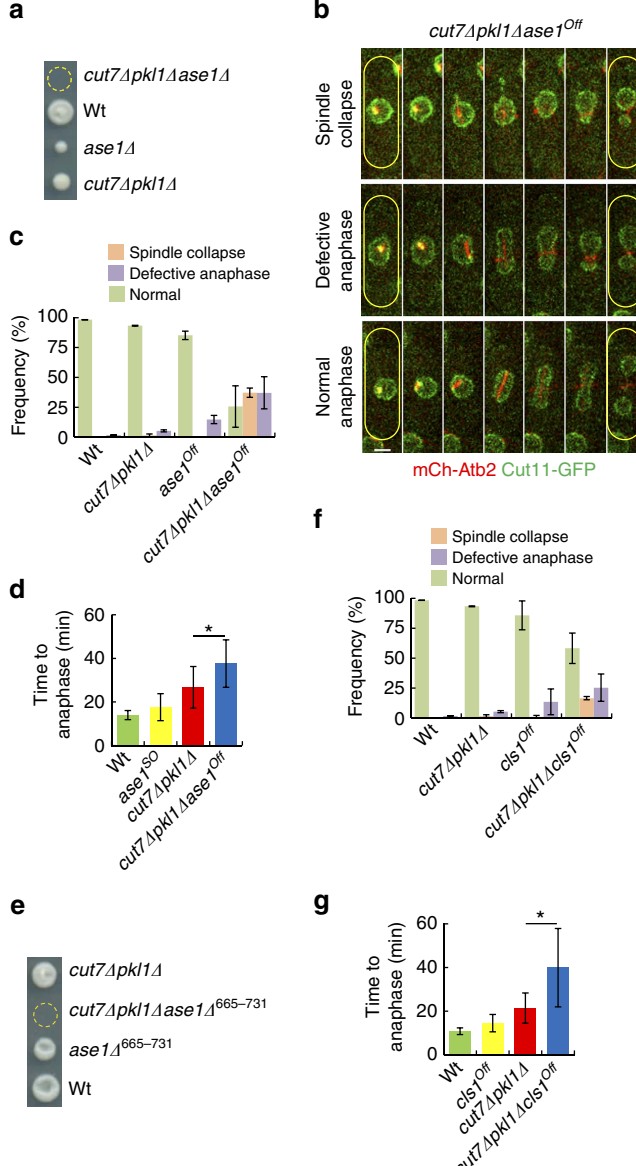

**Figure 3 | Ase1 and Cls1 are critical for spindle assembly in *cut7Δpkl1Δ* cells.** (**a**) Tetrad dissection of *cut7Δpkl1Δ* cells crossed to *ase1Δ* cells. (**b**) Time-lapse images of *cut7Δpkl1Δase1^Off^* mitotic cells expressing mCherry-Atb2 and Cut11-GFP (NE marker), after 20 h of repression/shut-off. Each frame corresponds to 5 min interval. Each row of images represents the detected phenotypes. (**c**) Bar plot of frequencies of spindle phenotypes in wild-type (*n* = 240), *cut7Δpkl1Δ* (*n* = 197), *ase1^Off^* (*n* = 123) and *cut7Δpkl1Δase1^Off^* cells (*n* = 88) after 20 h of repression/shut-off. (**d**) Bar plot showing the anaphase transition time of wild-type (green; *n* = 17), *ase1^Off^* (yellow; *n* = 23), *cut7Δpkl1Δ* (red; *n* = 25), *cut7Δpkl1Δase1^Off^* cells (blue; *n* = 21) after 20 h of repression/shut-off. *P < 0.001. (**e**) Tetrad dissection of *cut7Δpkl1Δ* cells crossed to *ase1Δ^665–731^* cells. (**f**) Bar plot of frequencies of spindle phenotypes in wild-type (*n* = 240), *cut7Δpkl1Δ* (*n* = 197), *cls1^Off^* (*n* = 109) and *cut7Δpkl1Δcls1^Off^* cells (*n* = 121) after 20 h of repression/shut-off. (**g**) Bar plot showing anaphase transition time of wild-type (green; *n* = 42), *cls1^Off^* (yellow; *n* = 33), *cut7Δpkl1Δ* (red; *n* = 29) and *cut7Δpkl1Δcls1^Off^* cells (blue; *n* = 28) after 20 h of repression/shut-off. *P < 0.001. Data show mean ± s.d. and Student's *t*-test *P* value. Scale bars, 2 μm.

after a pronounced delay compared to the *cut7Δpkl1Δ* control (Fig. 3d). Altogether, these data indicate that Ase1 contributes to bipolar spindle assembly in *cut7Δpkl1Δ* cells.

Ase1 also contributes to spindle stability by recruiting the CLASP protein Cls1 (ref. 20). CLASP proteins bind the MT lattice, suppressing MT catastrophe and promoting MT rescue, thereby contributing to overall MT stability[19,29]. To define the role of MT stability in bipolar spindle assembly in *cut7Δpkl1Δ* cells, we used a C-terminal truncated Ase1 (*Ase1Δ^665–731^*), incapable of properly recruiting Cls1 to the spindle. Tetrad dissection determined that *ase1Δ^665-731^* is lethal in *cut7Δpkl1Δ* cells (Fig. 3e). Similarly, we could not combine the *cls1^36^* thermo-sensitive mutant with the *cut7Δpkl1Δ* background (Supplementary Fig. 3e), suggesting that full Cls1 activity is required to sustain *cut7Δpkl1Δ* spindle function. To determine the role of Cls1 in *cut7Δpkl1Δ* spindle assembly, a *cls1* shut-off strain (*cut7Δpkl1Δcls1^On/Off^*) was created and spindle behaviour was analysed upon Cls1 shut-off. Although this strain did not completely repress Cls1 expression, as shown by the mild phenotype obtained on Cls1 depletion in a wild-type background, it nevertheless produced a synthetic phenotype when combined with the *cut7Δpkl1Δ* cells, with ∼15% spindle collapse events, and another 25% of cells with defective spindle elongation in anaphase (Fig. 3f). Moreover, depletion of Cls1 in *cut7Δpkl1Δ* cells resulted in a dramatic delay in anaphase transition (Fig. 3g). These data indicate that Cls1-mediated MT stabilization becomes critical for *cut7Δpkl1Δ* bipolar spindle assembly.

**Increased MT stability rescues *cut7Δpkl1Δ* spindle assembly.** The short metaphase spindle and long metaphase delay displayed by *cut7Δpkl1Δ* cells suggest that wild-type metaphase spindle length is optimal for proper chromosome bi-orientation. We reasoned that increasing the *cut7Δpkl1Δ* metaphase spindle length may lead to faster anaphase transition. Interestingly, Ase1 overexpression increased metaphase spindle length in both wild-type and *cut7Δpkl1Δ* cells (Fig. 4a–d, Supplementary Fig. 4a), and reduced the anaphase transition time in *cut7Δpkl1Δ* cells (Fig. 4c–e). As expected, Ase1 overexpression also reduced the velocity of anaphase spindle elongation in the wild-type cells (Supplementary Fig. 4a), and blocked spindle elongation in *cut7Δpkl1Δ* cells (Fig. 4a,b), consistent with a defective anaphase elongation in the absence of Cut7 (Fig. 1e). Indeed, Ase1 can function as a molecular brake when accumulated at MT overlap regions[30]. The longer metaphase spindle length observed upon Ase1 overexpression may reflect longer interpolar bundled MT pushing on the opposite spindle pole as they grow, supporting spindle pole separation in metaphase[31]. To verify this, we increased MT stability and polymerization by mild overexpression of Cls1 (ref. 20). Cls1 overexpression increased spindle length at metaphase (Fig. 4f,g), and induced a slightly faster transition to anaphase (Fig. 4h–j). Moreover, we found that Cls1 overexpression rescued the *cut7^24^* strain thermosensitivity (Supplementary Fig. 4b), suggesting that MT arrays with increased stability can resist the inward pulling forces produced by Pkl1. To test this by other means, we increased MT stability by deleting the protein kinase A Pka1, which was shown to down regulate Cls1 (ref. 32). As expected, *pka1Δ* partially rescued the metaphase spindle length (Supplementary Fig. 4e,f) and anaphase transition time of *cut7Δpkl1Δ* cells (Supplementary Fig. 4c,d,g). Furthermore, *pka1Δ* also partially rescued *cut7^24^* thermosensitivity (Supplementary Fig. 4h).

**Ase1 and Cls1 are sufficient for spindle assembly *in silico*.** Finally, we developed a computational model of spindle assembly

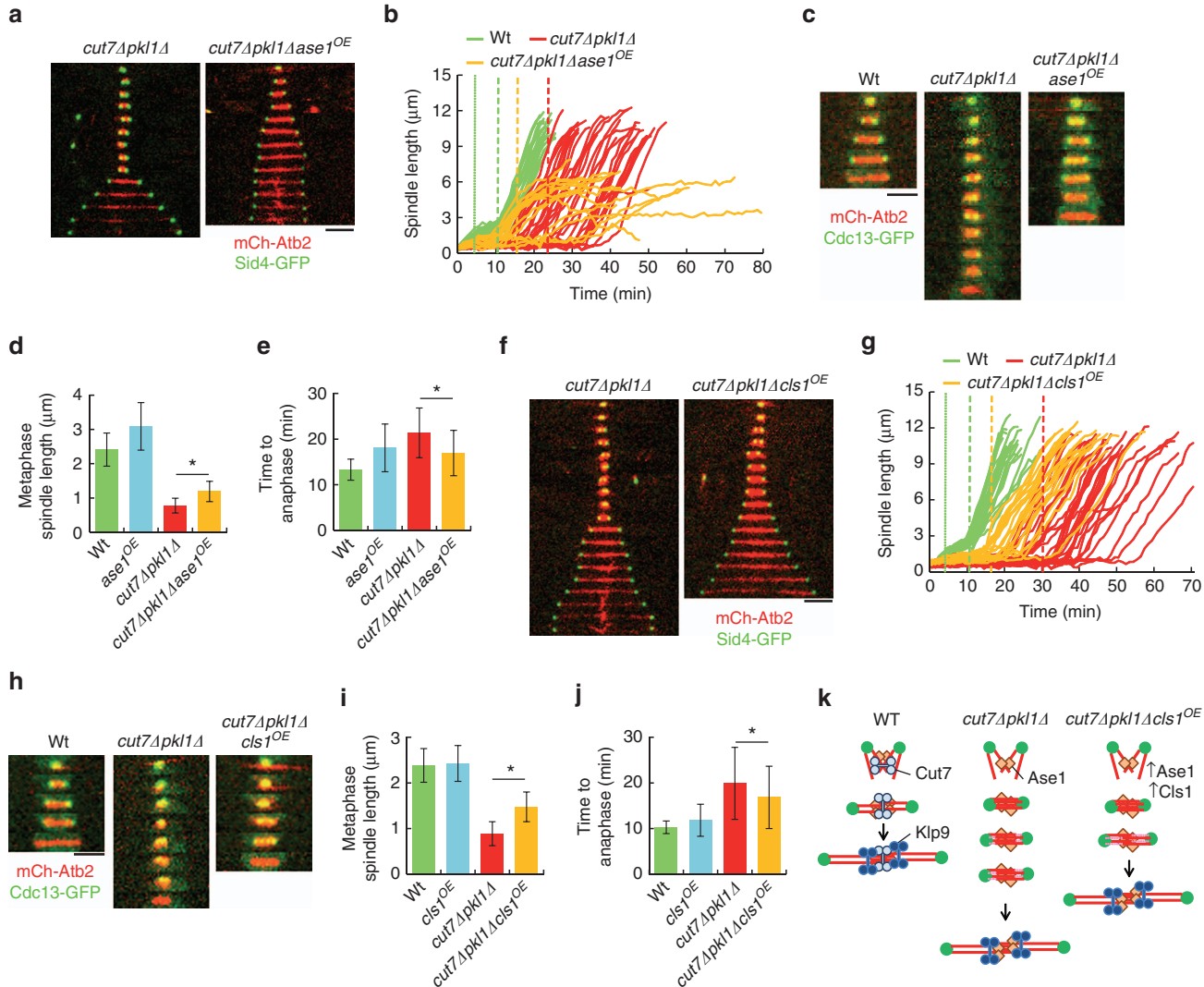

**Figure 4 | Increased metaphase spindle length reduces anaphase transition time in *cut7Δpkl1Δ* cells.** (**a**) Time-lapse images of *cut7Δpkl1Δ* and *cut7Δpkl1Δase1^OE* cells expressing mCherry-Atb2 and Sid4-GFP from mitosis onset to spindle breakdown. Each frame corresponds to 3 min interval. (**b**) Comparative plots of spindle length dynamics of *cut7Δpkl1Δ* (red curves; $n = 33$) and *cut7Δpkl1Δase1^OE* cells (orange curves; $n = 15$). Dotted vertical line corresponds to phase I–II transition; dashed vertical line corresponds to phase II–III transition. (**c**) Time-lapse images of wild-type, *cut7Δpkl1Δ* and *cut7Δpkl1Δase1^OE* cells expressing mCherry-Atb2 and Cdc13-GFP from mitosis onset to beginning of anaphase. Each frame corresponds to 3 min interval. (**d**) Bar plot showing metaphase spindle length at anaphase transition, reported by Cdc13-GFP degradation, of wild-type (green; $n = 33$), *ase1^OE* (light blue; $n = 28$), *cut7Δpkl1Δ* (red; $n = 25$) and *cut7Δpkl1Δase1^OE* (orange; $n = 30$). *$P < 0.001$. (**e**) Bar plot showing anaphase transition time of wild-type (green; $n = 33$), *ase1^OE* (light blue; $n = 28$), *cut7Δpkl1Δ* (red; $n = 25$) and *cut7Δpkl1Δase1^OE* (orange; $n = 30$). *$P < 0.01$. (**f**) Time-lapse images of *cut7Δpkl1Δ* and *cut7Δpkl1Δcls1^OE* cells expressing mCherry-Atb2 and Sid4-GFP from mitosis onset to spindle breakdown. Each frame corresponds to 3 min interval. (**g**) Comparative plot of spindle length dynamics of *cut7Δpkl1Δ* (red curves; $n = 31$) and *cut7Δpkl1Δcls1^OE* cells (orange curves; $n = 19$). Dotted vertical line corresponds to phase I–II transition; dashed vertical line corresponds to phase II–III transition. (**h**) Time-lapse images of wild-type, *cut7Δpkl1Δ* and *cut7Δpkl1Δcls1^OE* cells expressing mCherry-Atb2 and Cdc13-GFP from mitosis onset to beginning of anaphase. Each frame corresponds to 3 min interval. (**i**) Bar plot showing metaphase spindle length at anaphase transition, reported by Cdc13-GFP degradation, of wild-type (green; $n = 37$), *cls1^OE* (light blue; $n = 23$), *cut7Δpkl1Δ* (red; $n = 88$) and *cut7Δpkl1Δcls1^OE* (orange; $n = 97$). *$P < 0.001$. (**j**) Bar plot showing the time from MT nucleation to Cdc13-GFP degradation of wild-type (green; $n = 37$), *cls1^OE* (light blue; $n = 23$), *cut7Δpkl1Δ* (red; $n = 88$) and *cut7Δpkl1Δcls1^OE* cells (orange; $n = 97$). *$P < 0.05$. (**k**) Model for spindle assembly and transition to anaphase (see text for details). Data show mean ± s.d. and Student's *t*-test P value. Scale bars, 2 μm.

by Ase1 using a hybrid Brownian dynamics-kinetic Monte Carlo method (Methods: computational modelling), an extension of our previous work modelling spindle assembly[33]. In our model, Ase1 promotes MT bundling and crosslinks antiparallel-aligned MTs (refs 17,18,34). When bound to two MTs, Ase1 molecules exert spring-like forces (Fig. 5a). The NE is a sphere of fixed size with SPBs mobile in the envelope (Fig. 5b). Each SPB nucleates 14 MTs, which have minus-ends tethered to the SPBs (ref. 31; Fig. 5c). MTs are modelled as rigid rods that experience forces

from crosslinking Ase1 molecules, SPB tethering, random thermal forces and steric interactions with each other, and in some cases forces from MT plus ends contacting the NE. MTs undergo plus-end dynamic instability, characterized by growing and shrinking speeds and catastrophe and rescue frequencies. To model the stabilization of MT dynamics by Cls1 molecules recruited by Ase1 (ref. 20), MT dynamic instability is stabilized whenever an Ase1 is bound within a stabilization length of the end of an MT.

Because fission yeast cells unable to form stable kinetochore-MT attachments can still assemble bipolar spindles[35], we have neglected any mechanical contributions of chromosomes. Previous work has demonstrated that MT-kinetochore attachments and motors and MAPs at kinetochores affect spindle length, and can counteract effects of Ase1 (refs 28,36).

We chose to neglect kinetochores in chromosomes in our model to focus specifically on how Ase1 can organize MTs to establish bipolarity in the absence of motors. In future work, MT-kinetochore attachments and effects of motors and MAPs at kinetochores should be included in a full model of spindle force balance and length regulation.

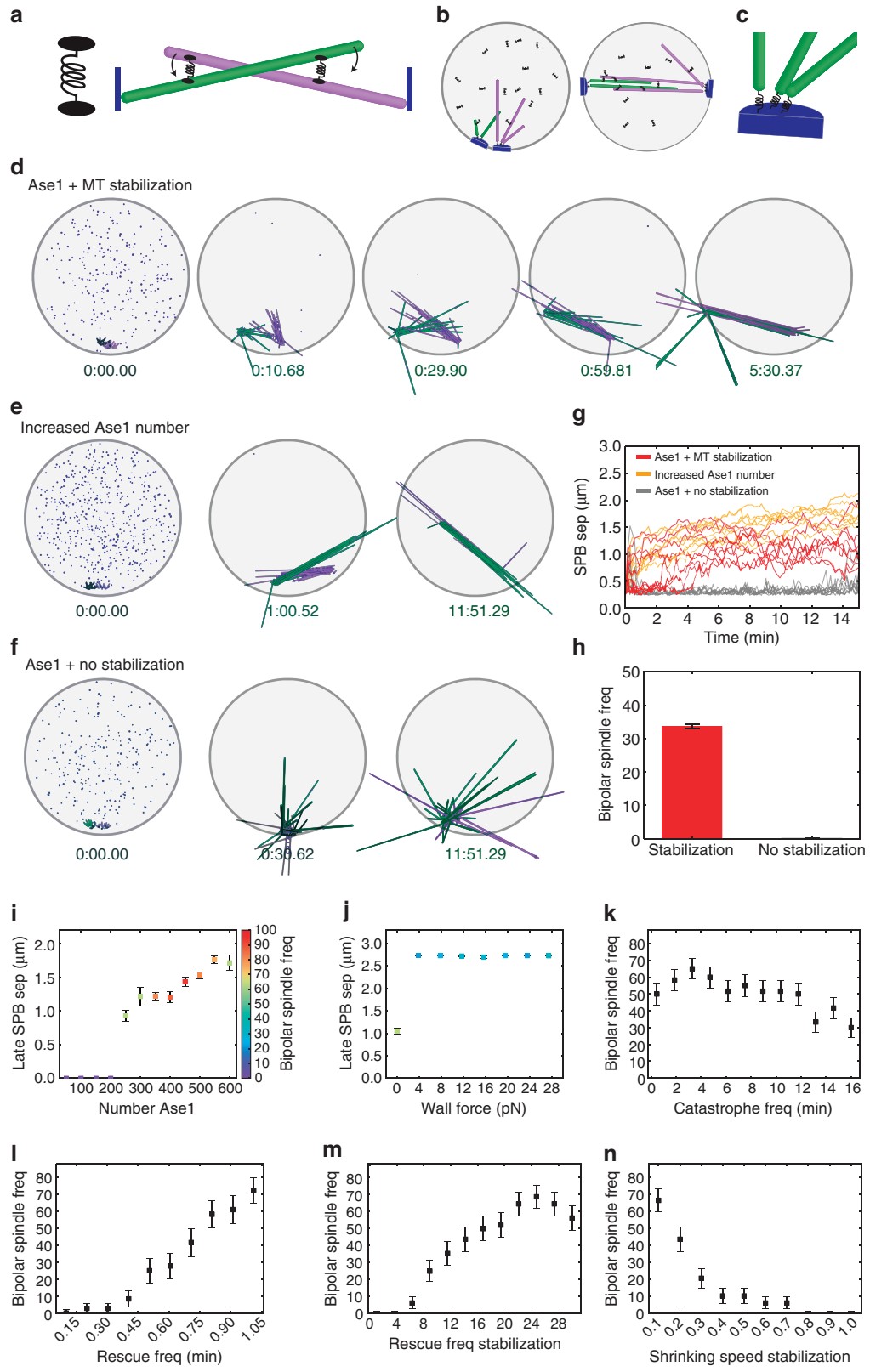

We found that with Ase1 molecules present and stabilization of MT dynamics to mimic the effects of Cls1, our model generated bipolar spindles from two side-by-side SPBs (Fig. 5d). The spindles are relatively short, consistent with our experimental results. As in the Ase1 overexpression experiments, increasing the number of Ase1 molecules in the spindle lengthens the spindle (Fig. 5e). Similarly, if we turn off the stabilization of MT dynamics by crosslinking, to mimic Cls1 deletion in the experiments, Ase1 can still crosslink MTs but spindle formation does not occur (Fig. 5f). To quantify these results, we determined the SPB separation as a function of time (Fig. 5g) for several different simulations for each parameter set used for panels d–f. These confirm the effects of increased Ase1 number on spindle length and the loss of spindle assembly in the simulated Cls1 deletion.

While these results compare well with our experiments for the reference parameter set (Supplementary Tables 1 and 2), several parameters in the model have not previously been directly measured experimentally. To understand the sensitivity of our model to parameter variation, we determined reasonable ranges for the poorly constrained parameters and performed simulations with parameter values randomly sampled over these ranges. We then quantified the fraction of simulations that were able to assemble a bipolar spindle, and compared that to another set of simulations with MT dynamics stabilization turned off to mimic Cls1 deletion (Fig. 5h). Bipolar spindles form in our model with random parameter sets and stabilized MT dynamics ∼30% of the time. Removing the dynamic stabilization to mimic Cls1 deletion abolishes bipolar spindle assembly.

We then studied the effect of individual parameters on spindle assembly and length, both by varying one parameter at a time starting from our reference values (Fig. 5i–n), and by analysing the influence of single parameters on the results of our random parameter sampling (Supplementary Figs 5–13). Both studies led to similar conclusions about the important parameters for spindle assembly driven by Ase1. Spindle length is sensitive to Ase1 number: for too few Ase1 molecules, bipolar spindle assembly does not occur, and above this minimum threshold increase Ase1 number leads to longer spindles (Fig. 5i, note that in panels i and j the colour of each point indicates the percentage of simulations for which bipolar spindles assemble, as shown in the colour bar).

Noteworthy, we also found that spindle length and assembly is quite sensitive to the 'wall force' experienced by MT plus ends that interact with the NE (Supplementary Computational Methods). Short spindles, with lengths similar to those in our experiments, occurred only when the wall force was set to zero (Fig. 5j). For any nonzero wall force, pushing forces from MTs interacting with the NE extended spindles, on average, to their maximum length with SPBs on opposite sides of the NE. Therefore, we found that the experimentally measured spindle lengths were only found in our model when the wall force was set

to zero. This likely occurs in our model due to the neglect of MT-kinetochore attachments, because the forces between sister kinetochores may exert inward forces that shorten the spindle, opposing pushing forces from MT-NE interactions. In our model that neglects MT-kinetochore interactions that lead to the pulling force, it appears that also removing the pushing force of MTs on the NE is necessary to achieve proper spindle length.

We found that in addition to MT number, the catastrophe frequency, rescue frequency, rescue frequency stabilization and shrinking speed stabilization have the largest impact on the establishment of bipolarity in our model. All of these are important because they affect the establishment or maintenance of antiparallel overlaps of MTs from opposite SPBs. We note that the values of these parameters may be constrained by the absence of chromosomes and kinetochores in our model, which if present might contribute to enhance stabilization of MTs from opposing SPBs. Therefore, the importance of stabilization of MT antiparallel overlaps in our model may be over emphasized with kinetochores absent. If the catastrophe frequency is either too low or too high, spindle assembly occurs less frequently (Fig. 5k). For low catastrophe frequency, this is because MTs are too long and turn over too slowly for antiparallel overlaps to form. For high catastrophe frequency, MTs are too short to successfully assemble a spindle. Similarly, if the rescue frequency is too low, bipolar spindles are unable to assemble because initial antiparallel MT contacts are lost when MTs shrink (Fig. 5l). Similarly, the rescue frequency must be highly stabilized by MT crosslinking, by at least a factor of 10, to maintain MT antiparallel overlaps for successful spindle bipolarity (Fig. 5m). The shrinking speed must be significantly decreased by crosslinking, by at least a factor of 5, for successful spindle assembly (Fig. 5n). These results highlight the importance of MT dynamics and its stabilization to formation of bipolar spindles by Ase1. Our reference parameter set which has a high frequency of spindle assembly has relatively short single MTs (mean length of ∼0.64 μm, Supplementary Computational Methods), because short MTs with rapid turnover allow initial antiparallel overlaps to be established quickly. Then the crosslink-induced stabilization moves the mean MT length into the unbounded growth regime[37], producing long, stable MTs that can maintain the bipolar spindle structure.

## Discussion

Proper spindle assembly requires the coordinated activities of motors and MAPs. Initial spindle pole separation is driven by kinesin-5, based on its unique properties of crosslinking and sliding antiparallel MTs apart[2–5,7]. Nonetheless, recent reports show that spindle bipolarity can be supported in the absence of kinesin-5, especially on depletion of inward forces produced by minus-end directed motors[8,9,11,13–16]. The kinesin-12 protein, Kif15, has emerged as an important player supporting spindle bipolarity on kinesin-5 inhibition in human cells[12,38–40].

**Figure 5 | Computer modelling reveals Ase1 essential role in spindle formation.** (**a**) Schematic of Ase1 effects. Ase1 (black) crosslinks antiparallel MTs (green, purple) and promotes MT alignment and bundling. (**b**) Model schematic. The NE (grey sphere) contains mobile but membrane-bound SPBs (blue). MTs (green, purple) are tethered to SPBs. Ase1 molecules (black) diffuse within the volume enclosed by the NE and can attach to and move along MTs. (**c**) Schematic of SPBs and MT minus ends. MTs are tethered via springs to SPBs. (**d**) Image sequences of spindle assembly simulation. Initially the SPBs are adjacent, and each MT is 0.1 μm long (t = 0). Black spheres/lines indicate unbound/bound Ase1 molecules, which initially diffuse in the nucleoplasm. Times shown are in minutes and seconds. A bipolar spindle forms, elongates and remains stable through the end of the 10-min simulation. (**e**) Image sequences of spindle assembly with increased Ase1 number. (**f**) Image sequences of spindle assembly simulation that mimics Cls1 deletion. Ase1 molecules are present and can crosslink antiparallel MTs, but Ase1 does not cause stabilization of MT dynamics. Spindle assembly does not occur. (**g**) Quantification of multiple simulations using the parameters of **d**–**f**, showing SPB separation as a function of time. (**h**) Frequency of bipolar spindle assembly for randomly sampled parameter sets with and without stabilization of MT dynamics by crosslinking. (**i**–**n**) Dependence of spindle assembly and length on single varied parameters. (**i**) Number of Ase1 molecules, (**j**) wall force, (**k**) catastrophe frequency, (**l**) rescue frequency, (**m**) rescue frequency stabilization, (**n**) shrinking speed stabilization. Note that in **i** and **j** the number of simulations corresponding to each point is indicated by the colour of the point, as shown in the colour bar. Data show mean ± s.d.

The mechanism behind this remains elusive, since kinesin-12 cannot produce extensive sliding of antiparallel MTs to separate centrosomes, but rather favors the reorganization of MTs into parallel bundles[41]. Interestingly, fission yeast can assemble a bipolar spindle in the absence of Cut7 and Pkl1 (refs 15,16) but does not contain kinesin-12 homologues, pointing towards yet other kinesin-5-independent pathways to promote spindle assembly. Indeed, we show that spindle bipolarity can be reached with absolutely no motor activity (Fig. 2i), but instead requires the antiparallel MT bundler Ase1 (Fig. 3c). Similar to budding yeast, Ase1 works in parallel with Cut7 to support SPB separation in fission yeast. Ase1 may promote spindle bipolarity and stabilize the spindle at initial spindle formation by crosslinking antiparallel MT (ref. 42). Further, Ase1, through inherent entropic expansion forces[34], may promote inward MT sliding, resulting in longer MT overlapping midzone region and more stable spindle. In addition, we show that a critical role for Ase1 in this context is to recruit Cls1 to promote MT stability. Interestingly, increasing MT stability also promoted the formation of longer metaphase spindles in cut7Δpkl1Δ cells, indicating that MT polymerization can push and sustain spindle pole separation at this stage, and can translate into a more efficient SAC satisfaction, accelerating the anaphase transition (Fig. 4b,g). Noteworthy, MT polymerization is key, because MT stabilization alone would not translate into force to push the poles apart. Brownian dynamics-kinetic Monte Carlo simulations confirmed that Ase1 and Cls1 activities are sufficient to form a bipolar spindle in the absence of motors (Fig. 5).

Kinesin-5 involvement in spindle dynamics spans from the initial spindle assembly to the end of anaphase, since we showed that it plays a role in anaphase elongation. While it has been suggested that kinesin-5 may act as a brake in C. elegans embryos[43], in other systems such as S. cerevisiae or Drosophila embryos, it is the major protein responsible for antiparallel MT sliding to elongate the spindle[44–46]. In the absence of kinesin-5 Cut7, the major outward force producer for spindle assembly, fission yeast kinesin-6 Klp9, which is involved in anaphase spindle elongation[23], could participate in earlier stages. However, our results show that Cut7 works in parallel with Klp9 to successfully elongate the spindle only during anaphase, but not before (Fig. 2c). Interestingly, while Klp9 recruitment to the spindle is controlled by Clp1-dependent de-phosphorylation at anaphase onset, Cut7 activity might be constant throughout mitosis, constituting a back-up mechanism to ensure sister chromatid segregation in anaphase.

Our work suggests a model (Fig. 4k), where in wild-type cells, MTs nucleated from SPBs at mitotic entry are crosslinked by Cut7 and Ase1 to promote spindle bipolarity and an optimum length at metaphase. Once chromosomes are bi-oriented, Klp9 recruitment promotes fast elongation of the spindle. In cut7Δpkl1Δ cells, Ase1 still promotes spindle bipolarity. Further, Ase1 recruits Cls1 which promotes MT stability.

In recent years, examples of spindle assembly in the absence of kinesin-5 have begun to emerge. While kinesin-5 inactivation results in spindle collapse in most cases, in rare occasions, cells can cope with it, by using alternative pathways. Our work shows that increased MT bundling and stability sustains spindle bipolarity in the absence of kinesin-5 in fission yeast. Further work will be important to identify additional proteins involved in this process and evaluate their contribution to spindle assembly, which undoubtedly will contribute to the design of novel anti-mitotic therapies.

## Methods

**Fission yeast genetics and culture.** Standard *Schizosaccharomyces pombe* media and genetic manipulations were used[47]. All strains used in the study were isogenic to wild-type 972 and are described in Supplementary Data 1. Strains from genetic crosses were selected by random spore germination and replica in plates with appropriate supplements or drugs. Transformations were performed using the lithium acetate–dimethyl sulfoxide method[48]. Briefly, 25 ml of exponentially growing cells were washed with tris-ethylenediaminetetraacetic acid (EDTA) buffer (10 mM Tris, 1 mM EDTA, pH 8) + 0.1 M lithium acetate, then concentrated to 1 ml via centrifugation. 1 μg of purified DNA was added to the cells, mixed with 2 μg of denatured single-stranded DNA, 10 μl of dimethyl sulfoxide and 700 μl of Tris-EDTA buffer + lithium acetate 0.1 M + 40% PEG, and incubated at 30 °C for 40 min. Cell mix was heat-shocked at 42 °C for 7 min, then cells were washed with water and plated onto plates containing the corresponding selective media.Generally, cells were cultured in YE5S incubated at 25 °C and imaged during exponential growth. Overexpression experiments were performed with exponentially growing cells incubated at 25 °C for 20 h in Edinburgh minimal medium (EMM) without thiamine, supplemented with adenine, uracil and leucine. For shut-off experiments, cells were cultured in EMM without thiamine, supplemented with adenine, uracil and leucine, and transferred to YE5S (which contains thiamine) for 8 (for klp9 shut-off) or 20 h (for ase1 and cls1 shut-off).

Tetrad dissection was performed YE5S plates using a Singer dissection microscope MSM 400. Plates were incubated 4–5 days at 25 °C before replicating onto the appropriate selection plates.

Serial dilution assays were performed on YE5S plates (except for Supplementary Fig. 4B, in which EMM plates supplemented with adenine, uracil and leucine were used), with a starting OD of 2.0, and fourfold dilution (except for Supplementary Fig. 4B, in which the starting OD was 4.0) and incubated 2–3 days at the specified temperature.

**Production of mutant and tagged strains.** To produce strains overexpressing *klp9*, *ase1*, *GFP-ase1*, *cls1* or *GFP-pkl1* from their endogenous loci using the thiamine repressible *nmt81*, *nmt41* or *nmt1* promoter[49], cells were transformed with KanR:nmt81/41/1 cassettes PCR-amplified from pFA6a plasmids[48], then selected on YE5S-G418 plates and transformants were confirmed by diagnostic PCR.

For Cut7-mCherry overexpression, *cut7-mCherry* was integrated into the *leu1* locus under the control of the *nmt41* promoter. A pJK148 vector[50] containing the *nmt41* promoter amplified from pFA6a-KanR-P41nmt1 (ref. 48), and subcloned between KpnI and SalI sites, *cut7* ORF amplified from genomic DNA purified from a wild-type strain and subcloned between SalI and EcoRI sites, mCherry amplified from pFA6a-mCherry-NatR and subcloned between EcoRI and NotI sites and *nmt1* terminator amplified from pAP146 (ref. 51) and subcloned between NotI and SacI sites, was digested with AfeI to circularize the plasmid and allow integration in the *leu1* locus of a wild-type strain.

Constructs expressing *sid4-GFP* or *cut11-GFP* were integrated into the *leu1* locus under the control of their own promoter. Briefly, fragments containing ~1 kb of 5′ UTR and the ORF of *sid4* or *cut11* were amplified from genomic DNA purified from a wild-type strain and cloned between KpnI and NotI sites of pAP146 (ref. 51). Plasmids were circularized by NruI digestion to allow integration in the *leu1* locus of a wild-type strain.

Constructs expressing *sid4-mCherry* were integrated into the *leu1* locus under the control of their own promoter. Briefly, a fragment containing ~1 kb of 5′ UTR and the ORF of *sid4* was amplified from pJK148 pSid4-Sid4-GFP and subcloned between KpnI and EcoRI sites of pJK148 pnmt41-Cut7-mCherry. The plasmid was circularized by NruI digestion to allow integration in the *leu1* locus of a wild-type strain.

For production of the cut7Δpkl1Δklp2Δklp6Δklp5Δ strain, TP3225 (cut7:NatR pkl1:KanR mCherry-atb2:HygR leu1:sid4-GFP) was sequentially deleted for *klp2*, *klp6* and *klp5*, using a *ura4* marker, which was deleted by transformation with a PCR product containing the 5′ and 3′ UTR of the correspondent gene and counter-selected for *ura-* phenotype on 5-FOA plates.

**Microscopy and image analysis.** Live-cell imaging was performed in micro-fabricated microfluidic flow chambers, based on polydimethylsiloxane (PDMS), previously constructed and described from our laboratory[52]. Briefly, the design for the flow chamber was drawn using the computer-assisted design software L-Edit (Tanner EDA). The design was then laser etched into a thin layer of chromium on a quartz plate, which served as a photomask (Microtronics). Next, SU8 negative photoresist was spin coated onto a silicon wafer (MicroChem). The features were then transferred from the photomask onto the photoresist layer by exposure and crosslinking with ultraviolet light (365 nm) for 20 s. The photoresist was developed with SU8 developer and cleaned with isopropyl alcohol and nitrogen gas. The wafer was then able to serve as a master mould on which repeated replication of moulds could be made by casting from PDMS (Dow Corning). Chambers were assembled by peeling off a PDMS replica from a mould, introducing inlet and outlet holes, and bonding the replica to a microscope glass coverslip after surface treatment using a plasma cleaner (Harrick Scientific).

Imaging was performed at room temperature (~22 °C). Images were taken on the motorized inverted Nikon Eclipse Ti-E microscope, controlled with MetaMorph 7 software and equipped with a spinning-disk CSUX1 confocal head (Yokogawa Electric Corporation), a Plan Apochromat × 100/1.45NA oil immersion objective lens (Nikon), a PIFOC objective stepper, a Mad City Lab piezo

stage, a CCD CoolSNAP HQ2 camera (Photometrics) and a laser bench (Errol) with 491 and 561 nm diode lasers, 100 mW each (Cobolt). 7 planes stack images were taken for each channel (usually 300 msec exposure time, binning 2, electronic gain 3, 12% laser power for GFP or 15% laser power for mCherry for each plane). Maximum projections obtained for presentation and analysis.

All time-lapse movies were 90–120 min long, taken with 1 min interval. Spindle dynamics graphs were obtained by analysis of spindle pole distance over the time, and drawn with Microsoft Excel 2010.

**Computational modelling.** Our model uses hybrid computational schemes that combine Brownian dynamics and kinetic Monte Carlo simulation. Brownian dynamics governs the motion of physical objects, such as MTs and SPBs, by incorporating both deterministic forces/torques due to steric interactions and motors/crosslinkers as well as random forces/torques due to thermal fluctuations. Kinetic Monte Carlo methods model stochastic events that change the state of molecules in the system, including motor/MAP binding/unbinding and MT dynamic instability. We have used these techniques to model MT-motor mixtures[47–49] and spindle assembly[33].

The model used here is an extension of that described in Blackwell et al.[33]. Briefly, the simulation takes place within a sphere of constant shape and diameter, which represents the NE of fission yeast. The SPBs are represented as thin disks inserted into the NE, each with 14 MT nucleation sites[31]. Each SPB can move (translate and rotate) within the NE due to forces exerted on it by the attached MTs, random thermal forces and drag forces from the envelope, which oppose SPB motion. We used measurements of the SPB diffusion coefficient determined by tracking SPB motion in fission-yeast cells treated with methyl-2-benzimidazolecarbamate (MBC)[33].

MTs are modelled as rigid rods which interact via nearly hard-core interactions. The MTs diffuse both rotationally and translationally, constrained by the tethering of their minus ends to the SPB. MT rotational diffusion about a pivot at the SPB was measured[50], allowing us to correctly compute MT diffusion for any length. MT plus-end dynamic instability is controlled by four parameters: the growth and shrinking speeds, and catastrophe and rescue frequencies. These parameters have been previously measured[50–52], but we found that modifications to these parameters were needed to allow the model to more robustly assemble bipolar spindles in our model, as discussed in the main text.

Our model includes the alteration of MT dynamics by Ase1. Ase1 binds preferentially to antiparallel MT overlaps[53,54] and then recruits Cls1 to stabilize MT dynamics[20]. We have modelled these effects by allowing preferential Ase1 crosslinking of antiparallel MT pairs only. To model Ase1 recruitment of Cls1 and the subsequent stabilization of MT dynamics, we adjust the dynamic instability parameters when a crosslink is within the threshold distance $s$ of an MT's plus end. Each dynamic instability parameter for the attached MT is scaled:

$$
\begin{aligned}
f_c &= f_{c,0} S_{f_c} \\
f_r &= f_{r,0} S_{f_r} \\
v_g &= v_{g,0} S_{v_g} \\
v_s &= v_{s,0} S_{v_s}
\end{aligned}
\qquad (1)
$$

We compute the full filament pair partition function to ensure that crosslinker kinetics follow the correct statistical mechanical rules[33,48,55]. Binding and unbinding occur for either head individually, so the crosslinkers can have zero, one or two heads bound. The binding and unbinding rates are chosen to ensure that the correct equilibrium distribution of bound crosslinkers is recovered for static crosslinkers, and to allow force-dependent binding kinetics. Unbound crosslinkers freely diffuse in the nucleoplasm. The crosslinkers do not sterically interact with each other, either while free or bound. Bound crosslinkers exert forces as harmonic springs if they stretch/compress away from their rest length. Bound crosslinkers exert forces and torques on MTs to which they are bound. Both bound heads of Ase1 diffuse along MTs in a force-dependent manner. Here we discuss extensions to the previously published model[33].

To determine the effects of parameter variation, we conducted random sampling of 9 parameters: the number of Ase1 molecules, the MT growth and shrinking speeds and rescue and catastrophe frequencies, and the stabilization factors contributed by the crosslinkers to MT dynamics. The ranges of these parameters are in Supplementary Tables 1 and 2. 500 different parameter sets were created by uniformly randomly sampling each parameter over its allowed range.

For each parameter set, we performed 12 different simulations and determined the fraction of successful bipolar spindles formed. To identify successful spindle assembly, we measured the interpolar fraction (IF) of spindle MTs[33]. A spindle was considered successful if it had an IF > 0.2 for longer than 2 min and lasting until the end of the simulation. We chose this criterion after observing many simulations, which showed that if the IF remained this high for at least 2 min, then the bipolar spindle was typically stable to the end of the simulation. Because the IF can fluctuate for short periods, spindles for which the IF dropped below 0.2 for up to 12 s within this 2 min time period were still considered successful. This criterion proved effective in filtering long-lasting bipolar spindles from transient or monopolar spindles.

If a bipolar spindle successfully formed, the late-time SPB separation was calculated by averaging the distance between SPBs for the last 30 s of the

simulation. We also measured the average MT length and spindle formation start time. The start time was defined as the time at which a successful spindle first crossed the 0.2 IF threshold.

A second random parameter sampling run was performed with all stabilization factors held at one to model no stabilization, as would occur in a Cls1 deletion mutant. The other five varied parameters were allowed to sample the parameter space as described above. The smaller parameter space meant that we required fewer runs to achieve reasonable statistics; therefore, only 200 parameter sets were simulated. Again, for each parameter set we performed 12 different individual simulations. Only 2 out of the 2,400 simulations showed successful spindle formation by our criteria.

After observing trends in the random parameter sampling data, we systematically tested parameters that showed the greatest effect on spindle success and late-time SPB separation. First, we selected values of the 9 parameters that we had varied in our random parameter sampling scan that best correlated with either previously measured values or, if unavailable, gave a high rate of bipolar spindle success. We chose 24 linearly spaced values for asymptotic wall force between 0–14.8 pN and ran 32 individual simulations for each of these values. Of these, only the zero wall force parameter set showed dramatically reduced late-time SPB separation which agreed with experimentally observed measurements of spindle length. We then individually varied the number of crosslinkers, the rescue and catastrophe frequencies, the rescue frequency stabilization, and the shrinking speed stabilization. All other parameters were held constant at our reference values (Supplementary Tables 1 and 2). We used 10–12 linearly spaced values in the ranges given in Supplementary Tables 1 and 2.

In modelling the force that would be exerted on MTs by the deformation of the NE, we used the model of membrane tube extension studied previously[56]. To approximate the force generated normally inward by the spherical boundary, we broke the wall force into the linear and non-monotonic regimes. In the linear regime we used the equation from Derenyi et al.[56].

$$
F_{lin}(L) = \frac{F_w}{R_{tube}(\ln(2) - \gamma)} L
\qquad (2)
$$

where $L$ is the distance the MT protrudes from the wall, $F_w$ is the asymptotic wall force, $\gamma = 0.577...$ is Euler's constant and $R_{tube}$ is the characteristic membrane tube radius. $R_{tube}$ can be calculated from the bending rigidity of the membrane and the membranes surface tension[57]. However, this linear equation quickly diverges from the true force. For larger distortions, we approximate the force by:

$$
F(L)_{asmp} = 2aF_w e^{\frac{-L}{b}} \cos\left(\frac{L}{b} + c\right) + F_w
\qquad (3)
$$

where the integration constant $a = 0.5416...$, $b$ is $\sqrt{2R_{tube}}$, and $c = 4.038....$ This expression cannot be used for lower values of $L$ because the force does not tend to zero as $L$ approaches zero. By adding these two equations and multiplying the non-monotonic term by a factor of $1 - e^{-L}$ the correct boundary condition at $F(L=0)=0$ and asymptotic response were achieved with a single equation. This force only effects the plus tip of the MT and is always directed radially inward.

Our model also incorporates the force-dependence of MT growth speed[58]. To model this, we adopted the multi-filament Brownian ratchet model described by van Doorn et al.[59]. Assuming each protofilament of a 13-protofilament MT increases the MT length by an average of $\sigma = 8/13$ nm and each has an equal chance of polymerization, the growth speed becomes

$$
v(F) = v_+ \left( N(e^{-N\beta} - \alpha^N) + (1-\alpha) \sum_{n=1}^{N-1} n(e^{-n\beta} - \alpha^n) \right)
\qquad (4)
$$

where we have defined $\alpha = e^{F\sigma/kT}$ and $\beta = F\sigma/kT$. The number of protofilaments $N = 13$. The constant $v_+$ is:

$$
v_+ = \frac{v_{p,0}}{\left( N(1-\alpha^N) + (1-\alpha) \sum_{n=1}^{N-1} n(1-\alpha^n) \right)}
\qquad (5)
$$

and the force acting on the MT's tip, $F$, is the component of the force exerted by the boundary acting along the MT axis.

**Data availability.** The authors declare that all data supporting the findings of this study are available from the corresponding authors upon request.

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

## Acknowledgements

We thank Matthew Glaser, Loren Hough and Dick McIntosh for useful discussions. This work was supported by National Science Foundation grant no. DMR-1551095 and National Institutes of Health K25 GM110486 and R01 GM104976 to MB, a fellowship to A.L. provided by matching funds from the National Institutes of Health/University of Colorado Biophysics Training Program, and facilities of the Soft Materials Research Center under the National Science Foundation Materials Research Science and Engineering Centers grant DMR-1420736. We acknowledge the Biofrontiers Computing Core at the University of Colorado Boulder for providing High Performance Computing resources (NIH 1S10OD012300) supported by BioFrontiers IT. We thank the labs of Fred Chang (UCSF), Sophie Martin (University of Lausanne) and Ian Hagan (University of Manchester) for generously providing strains. The imaging were performed on PICT-IBiSA (Institut Curie), a member of the France-BioImaging national research infrastructure. We thank Imen Bouhlel and Kathleen Scheffler for advice and discussions.

This work is supported by the Agence Nationale de la Recherche, Fondation ARC pour la Recherche sur le Cancer, La Ligue Contre le Cancer, Institut National du Cancer and the National Institutes of Health to PTT. The Tran lab is a member of the Labex CelTisPhyBio, part of IdEx PSL.

## Author contributions

A.L., R.B. and M.D.B. designed, implemented and wrote up the Brownian dynamics-kinetic Monte Carlo simulation. S.A.R. designed and performed fission yeast experiments, analysed data and wrote the paper; V.S. initiated the project; V.F. assembled and maintained the microscopes; M.D.B., A.P. and P.T.T. edited the paper.

## Additional information

**Competing interests:** The authors declare no competing financial interests.

