## [Peer Review File · Nature Communications]

Reviewer #1 (Remarks to the Author) .

The authors have responded to my original concerns...

1. in Figure 5 the word stabilization has a ?, and this needs to be fixed...

2. In figure 2D the red curves obscure the blue ones; perhaps change the colors? (for example 4C is easier to see all three)...

Reviewer #2 (Remarks to the Author) .

In the revised manuscript, the authors have addressed my concerns and questions. I think that this is an important study that significantly impacts our understanding of functional interactions between mitotic motors, microtubule crosslinkers and MT polymerases during the assembly and function of the mitotic spindle. It should be of interest to the journal readership and accordingly I recommend its publication in Nature Communications...

Reviewer #3 (Remarks to the Author) .

The authors have clarified my experimental concerns and some of the simulation concerns. Unfortunately I still have some concerns on the conclusions drawn from the simulations (see below). Overall, the message of the paper is important for the community and it is proven by the experiments, which are very sound. Therefore, I feel the paper should be published in Nature communications, but I think some issues with the simulations have to be clarified...

My major concern is the claim of necessity on MT stability and antiparallel bundling by ASE1 to form spindles in absence of motors (which is the header of a section of the paper). This claim is simply not justified with the work provided for several reasons. The foremost and most important is that the kinetochore-MT interactions are not considered. I think it is fine to leave that for future work as the authors argue, but since the possibility that this interaction may generate bipolar spindles in the absence of motors is not tested, then one cannot claim necessity. The justification in the text "Because fission yeast cells unable to form stable kinetochore-MT attachments can still assemble bipolar spindles 35, we have neglected any mechanical contributions of chromosomes." sounds reasonable when trying to check sufficiency, but not necessity, as in the assembly of these spindles the motors were present. The fact that the simulation is consistent with the experimental results is already very interesting and valuable, and I think the results as they stand cannot be pushed further than this...

Another issue I had during the previous revision was the unreasonably long length of the simulated spindles (and the choice of the dynamic parameters). This issue is not really resolved satisfactorily. For instance, the authors now write that in the presence of the wall force the spindles elongate to this unrealistic length, and only when the nuclear envelope force is set to 0 the spindles are short. To me that seems an inconsistency of the model because why wouldn't the microtubules not feel the envelope? At least a justification other than stating that without setting this force to 0 the right length cannot be obtained should be added. Arguing that MT-kinetochores may be the reason for the shortening the spindle argues for simulating the MT-kinetochores. The dynamic parameters for MTs lead to an unbound growth (as the authors have now added to the text), but since the force from the nuclear envelope is set to 0 now, does this mean the MTs have an unbound average length? Isn't the wall force necessary to trigger MT depolymerization? .

..The dynamics parameters have been modified from literature "to increase the success of bipolar spindle assembly with Ase1 only" (quoted from their response). I find that modifying the parameters to obtain the expected result is biasing the conclusions from the simulations. Adding to this, the justification to not use more reasonable parameters (that would lead to a reasonable MT length) because otherwise "problems with Ase1-only spindle assembly occur" is less than satisfactory: isn't the point of the simulation to show precisely that? perhaps the fact that

problems with spindle assembly occur using measured parameters points to more fundamental problems of the model..

..In summary, while the simulation is consistent with the assembly of bipolar spindles in the absence of motors, I find there are major issues in the simulations that are not consistent with the data. Therefore, I think it is necessary to tone down the conclusions drawn from the simulations and state only sufficiency or consistency..

Minor:..

..-Why is the criterium for a spindle to be viable to have an IF greater than 0.2 for longer than 2 minutes? And why IF is allowed to drop below 0.2 for 12 seconds? all these criteria sound a bit arbitrary and at least a rationale should be added..

- in the supplement there are many typos. For instance (and I stopped writing them down after these) in Table S1, asymptotic wall force and membrane tube radius seem to be lacking the proper reference. The second paragraph of the same page "Each the results of each". The last paragraph of section "parameter shotgun tests" on page 16 is incomplete..

Reviewer #4 (Remarks to the Author)..

The authors have carefully and extensively addressed the concerns of the referees, and the manuscript has clearly improved. I recommend publication in Nature Communications..

Rebuttal to the Reviewers

We thank the four Reviewers for their time and effort in reading our manuscript, and for suggestions to make our paper even better.

In the revised version we have addressed all concerns, and implemented all suggestions from the Reviewers. Briefly, we have performed additional fission yeast experiments and made modifications to Figs. 1, 2, 4 and Supplemental Figs. 2, 3, 4. Importantly, we have expanded the scope of the computer simulations, and included many additional data panels to Fig. 5. All text changes are highlighted in grey.

Below, we address specific points raised by the Reviewers.

Reviewer #1

In this manuscript the authors describe the results of experiments designed to determine how spindle bipolarity is established following depletion of the major outward and inward force generating kinesins. Using the fission yeast as a model, the results demonstrate that spindle formation and chromosome segregation occur in the absence of kinesin 5 (Cut7) and kinesin 14 (Pkl1), albeit on short spindles that show a delay in the time needed to enter anaphase. Under these conditions, the MAP Ase1 is essential for spindle bipolarity. They further show that fission yeast can undergo chromosome segregation when all kinesin motors, except for kinesin 6 to drive spindle elongation, are absent. The data emphasize the important role of microtubule dynamics in generating force for spindle formation in this organism. Overall, the manuscript provides new insight into the role of the microtubule associated protein, Ase1, in driving spindle formation.

1. The authors state that spindle elongation has three phases but these are not marked on the graph or images in Fig.1. This would be helpful. Additionally, on some of the later figures, it would be beneficial if a line representing the average of the wild type elongation kinetics could be added to show the difference between the particular mutant and the wild type condition.

As suggested, we have now drawn lines representing the average transition time between mitotic phases. Due to the inability to distinguish between phase I and II in the *cut7Δpkl1Δ* cells, only the transition between II and III has been added throughout the figures.

2. If the kinesin 14, which is proposed to antagonize Cut7 by generating inward forces, is localized at spindle poles (i.e. in the GFP-kinesin 14 images in the supplemental figure), how do the authors envision it works - pulling on minus ends near SPB? or is the motor acting between microtubules within the spindle?

We have clarified in the introduction the mechanism of kinesin-14 inward pulling forces. In previous studies from our lab and others (Syrovatkina and Tran, 2015; Yukawa et al., 2015), it was found that Pkl1 mostly localizes to the spindle pole when expressed under its own promoter, although a minor fraction of the protein can be detected on the spindle MTs. The spindle MTs localization becomes more evident under over-expressing conditions (Fig. S1). This result suggests that Pkl1 function is mainly restricted to the spindle pole, where its tail domain is anchored to the pole, and its motor domain may interact with plus-end MTs emanating from the opposite pole, and thus pulls the two poles together.

This seems to be consistent with recent data in budding yeast, which shows that increased levels of the kinesin-14 Kar3 at the spindle poles produced by the absence of Cik1 result in shorter metaphase spindle lengths (Hepperla et al., 2014), again pointing towards a model in which spindle pole-

anchored kinesin-14 produces inward pulling forces.

3. In the model, the force from kinetochore microtubule attachments (i.e. Dam1 and Klp5/6) is not included and the text states this is because the cells can establish bipolarity without kinetochore-microtubule attachments. Yet prior work showed that motors and maps at the kinetochore can antagonize Ase1. This needs to be addressed and clarified.

We agree with the Reviewer that this is an important point. In our opinion, MT-kinetochore attachments and effects of motors and MAPs at kinetochores should be included in a full model of spindle force balance and length regulation. We made the decision not to include them in this paper, because we wanted to focus specifically on how Ase1 can organize MTs to establish bipolarity in the absence of motors. Therefore we tried to include the smallest number of model ingredients that were able to produce this effect. We propose to further examine the effects of kinetochore attachments in future work. We have expanded discussion of this point in the paper.

4. The authors argue that Ase1 and Cls1 function by crosslinking and stabilizing microtubules to promote bipolarity. They show that overexpression of Cls1 rescues *cut724 ts* and ascribe this to microtubules with increased stability that "resist inward pulling forces". Growing microtubules generate force, and therefore it seems that microtubule polymerization, not just stability, is required. The authors seem to avoid any mention of the force of polymerization and emphasize microtubule stability; this point deserves more attention.

We agree with the Reviewer. This has now been stated properly in the discussion. Indeed, MT stability per se cannot provide any force to separate the spindle poles unless this is coupled to MT polymerization. As a side note, our unpublished results show that the MT polymerase Alp14/chTOG becomes essential in the *cut7Δpk11Δ* background, indicating that MT polymerization becomes a major force for spindle bipolarity.

In new simulations, we have further expanded on the interesting observation that the way that we model the effects of MTs interacting with the nuclear envelope has a significant effect on spindle length in our model. If MTs experience a force when their plus ends encounter the nuclear envelope, the polymerization forces make the spindle longer. Only when we turn off this "wall force" so that MT plus ends can encounter the nuclear envelope without experiencing a force are we able to get the short spindles that match the experimental measurements. We have further discussed this point in the revised manuscript.

5. On page 4 the authors test the contribution of the SAC to the delay in anaphase onset in the *Cut7 Pk11* double depletion strain. The data shows that the SAC is needed to delay anaphase onset so that proper connections can form. Their alternative hypothesis, that the spindle can't elongate due to the lack of *Cut7*, doesn't seem consistent with a delay in elongation.

We agree with the Reviewer. We have now removed the inconsis removing the first possibility to make easier the reading.

Minor:

On page 4 the text refers to figure 2 F,G, and I believe this should read figure 2 E,F.

Corrected.

Reviewer #2:

This is an interesting paper that addresses the mechanism of mitosis in fission yeast and the role of multiple redundant force generating mechanisms in spindle assembly and anaphase elongation. Using genetics, imaging and modeling, the authors examine mitosis in strains in which the contributions of various motors and MAPs have been modified, leading them to conclude that compressive forces generated by MT assembly can assemble bipolar spindles in the complete absence of mitotic kinesin motors, that kinesin-5 contributes to spindle elongation driven mainly by kinesin-6 and that Ase1p contributes to spindle assembly (i.e. prior to anaphase B onset) by recruiting the MT polymerase CLASP. I think these are all interesting findings that merit publication in Nature Communications after the following points are addressed.

1. It is unclear to this reviewer why the authors dismiss the possibility that Ase1p is required for bipolar spindle assembly because of its entropic force-generating mechanism (see Lansky et al, 2015, cited ref 35) rather than enhancing MT polymerization via the recruitment of CLASP. This requires better justification.

The Reviewer raises a good point. In the revision, we have discussed potential roles of Ase1 entropic forces. In deed, entropic force contributes to spindle bipolarity, particularly in early stages of spindle assembly, because Ase1 binding favors the creation of larger MT antiparallel overlaps. However, we argue that it is not the sole contributor for two reasons. First, both in the experiments and our model, removing the MT-stabilizing activity of CLASP abolishes spindle assembly. In the revised manuscript we have demonstrated that this effect holds over a wide range of model parameters, so that it appears that this stabilization is necessary for spindle bipolarity in our model. Therefore, the MT-stabilizing activity of CLASP appears essential. Second, the entropic force-generating mechanism by itself would predict that increasing Ase1 number via overexpression would shorten the spindle, because the larger number of crosslinkers favors greater spindle MT overlap through SPBs that are closer together. Both in our experiments and model, increasing Ase1 number lengthens the spindle, which can be explained through the greater stabilization of MT dynamics.

2. Fig 1E shows that anaphase spindle elongation slows down in cells deleted of kinesin-5 and kinesin-14 compared to wt, suggesting that "Cut7 (aka kinesin-5) participates in anaphase spindle elongation..." but how do we know that kinesin-14 deletion is not contributing to the slow-down? Again a better justification would be useful.

We performed new experiments to examine the possible role of kinesin-14 Pkl1 in anaphase spindle elongation. We did not detect significant differences in anaphase spindle elongation between wild-type and *pk11Δ* cells (Fig. S2D, S2F). Further, we did not detect significant differences in anaphase spindle elongation between *k1p9Δ* (Klp9 is the major anaphase spindle elongation motor (Fu et al., 2009)) and *pk11Δk1p9Δ* mutants (Fig. S2E, S2F). Taken together, we do not think Pkl1 contributes to anaphase spindle elongation.

3. On p4. the authors state that Klp9 functions as a homotetramer....to the best of my knowledge, this is based only on chemical crosslinking published in Fu et al (where on the gels presented, dimers appear to be the dominant species). This requires better evidence - especially useful would be the hydrodynamic techniques of Siegel and Monty.

We agree with the Reviewer that there is few functional data in the literature showing Klp9 working as a homo-tetramer. We have accordingly removed this statement from the revised text.

4. In the figures e.g. 1B, 2D etc it would be useful to mark phases I, II and III on the plots of spindle length versus time.

As suggested, we have now drawn lines representing the average transition time between mitotic phases. Due to the inability to distinguish between phase I and II in the *cut7Δpk11Δ* cells, only the transition between II and III has been added throughout the figures.

Reviewer #3:

In this paper, Rincon et al. study spindle formation in the absence of kinesin-5 in fission yeast. The authors find that in the absence of molecular motors, spindles can still form only if the microtubule antiparallel bundler PRC1/Ase1 is present to recruit CLASP/Cls1 that stabilizes microtubules. The authors conclude that pushing forces from polymerization are sufficient to assemble a bipolar spindle in the absence of molecular motors in fission yeast. The study is also complemented by simulations which are consistent with the data. The study addresses a fundamental question in the field of mitosis, that is what are the minimal requirements to assemble a bipolar spindle. The findings are interesting in that no motors are required to assemble a spindle but only an antiparallel bundler that stabilizes microtubules via the recruitment of a microtubule stabilizing MAP. The scope of this study is perhaps a bit narrow in that its conclusions may not apply and generalize to other bigger spindles, as yeast spindles are specially small and consist of typically only ~30 microtubules. The experimental part of the paper is convincing and the conclusions justified from the data itself alone. I have major concerns with the simulation part, however, which I find a bit weak and the conclusions from it a bit too strong (Ase1 and Cls1 being necessary and sufficient for spindle formation), see below.

Major comments:

The authors call the action of Ase1 + Cls1 leading to bipolar formation “pushing forces from MT polymerization”, but, where are the pushing forces actually acting on? Wouldn't it be more appropriate to call it expansion forces as, if I understood properly, are the entropic Ase1 forces that ultimately tend to increase the overlapping region of the stabilized MTs (which indeed keep growing as Ase1 recruits Cls1, leading to an increase of the overlapping region by Ase1)? Isn't it simply that the overlapping zone is proportional to the MT length, so Ase1 is ultimately exerting the force?

We agree with the Reviewer. We have now addressed the potential role of Ase1 entropic force in the discussion. We believe that Ase1 alone, i.e., entropic force alone, can indeed increase the overlapping MT midzone; but without new MT polymerization, the entropic force would bring the spindle poles closer together. Our data support this idea. The Ase1 truncation unable to properly recruit Cls1 to the spindle (Bratman et al., 2007), but keeps the MT binding domains, is not sufficient for spindle assembly (Fig. 3), suggesting that Cls1-dependent MT stabilization and subsequent MT polymerization is critical in *cut7Δpk11Δ* spindle assembly. This is further supported by the partial rescue of the defective spindle assembly in *cut7Δpk11Δ* cells by Cls1 over-expression.

In addition, our new simulations indicate that pushing forces from MT polymerization can promote spindle elongation when MT plus ends interact with the SPBs or the nuclear envelope. We have studied the effects of the “wall force” in our model, which describes the forces MT plus ends experience when they encounter the nuclear envelope. In our original model, the nuclear envelope was modeled as a hard wall. To model this more realistically and study its variation, we approximated the forces that MTs would experience upon distorting the nuclear envelope as they grow, and also studied the effects of eliminating this wall force (so that MTs can grow past the nuclear envelope radius without experiencing a force). We found that when a wall force of any magnitude is present, pushing forces contribute to spindle elongation. Only when the wall force was zero were we able to reproduce in the model the short spindles found experimentally. Therefore, we conclude that the “pushing forces” on MTs on the nuclear envelope and SPBs are not strictly necessarily. Thus, the

entropic expansion force of Ase1 is important for producing and enhancing MT antiparallel overlaps. However, we note that Ase1 recruitment of CLASP to stabilize overlapping MTs is important.

The experiments with the mutant lacking all mitotic kinesins (except Klp9, which is only involved in anaphase spindle elongation) are really informative and clearly show bipolar spindles can form without motors. The authors next dissect the role of antiparallel bundling and MT stability to finally show that these two activities are responsible for the bipolarity of the spindle. The authors do that using their previous *cut7 pkl1* mutant. To rigorously show that bundling and stabilization alone are responsible for bipolarity, shouldn't the authors use the mutant lacking all mitotic kinesins? Only then these experiments would certainly be a clean proof of Ase1 and Cls1 alone being responsible for spindle bipolarity in absence of motors.

We feel that we have shown conclusively, to the extent of our technical ability, that no motors are required for initial bipolar spindle formation (Fig. 2). The fact that Ase1 or Cls1 become essential in *cut7Δpkl1Δ* cells, indicates that the activity of other motors in this background are not sufficient to organize a bipolar spindle. Since the behavior of spindle dynamics in the *cut7Δpkl1Δ* strain is similar to that shown by the cells lacking all mitotic kinesins, we conclude that the contribution Ase1 and Cls1 to bipolar spindle formation in the absence of kinesin-5 is essential, while other mitotic motors play no measurable roles.

From the simulations, the authors state that Ase and Cls1 are necessary and sufficient for bipolar spindle formation. The sufficiency claim is justified as the simulation of an antiparallel bundler that drives MT stabilization does lead to a bipolar spindle. The necessity claim comes from the failure of obtaining a bipolar spindle in the absence of either antiparallel bundling or MT stabilization. I find some major issues with this conclusion and the simulation in general:

-These results are obtained for a particular set of parameters (12 of them are "estimated" without further justification) that are not measured nor referenced from the literature. Therefore claiming necessity requires extensively exploring these 12 free parameter space and show bipolar spindles cannot form in any condition. In fact, some of these parameters seem a bit unreasonable (see below).

We agree with the Reviewer, that our previous simulations with only one parameter set limited our conclusions. We have now extensively varied the parameter sets over wide ranges, for hundreds of different parameter sets, both with and without MT stabilization by crosslinking. Our conclusions that stabilization is an essential ingredient hold in this extended study.

-The authors neglect kinetochore-MT attachments and their mechanical contribution on the basis that bipolar spindles can form in cells unable for form stable kinetochore-MT attachments. However, since the authors are trying to find bipolar assembly in the absence of motors they should test whether kinetochore-MT attachments can lead to bipolar spindles in the presence of either a bundler or a MT stabilizer alone.

We agree with the Reviewer, that this is an interesting point to consider in the model. However, we chose in this work to focus on the minimal ingredients necessary to organize MTs into a spindle-like bipolar array, and chose to neglect MT-kinetochore attachments because they add significant additional complexity to the model and a large number of additional parameters. Model additional MT-kinetochore attachment is beyond the scope of the current work. We noted in the revision that the role of MT-kinetochore attachment will need future analysis.

- The values for the estimated parameters (i) need further justification, how were they fixed? In fact, these parameters are way off from reference values found in the literature and lead to an average MT length of $\sim 0,26 \mu\text{m}$ for the base parameters (see Verde et al., 92 for the expression of the average MT length), which seems really small. Why not using, as an estimate of MT dynamics parameters, the values found in yeast interphase MTs measured by some of the authors in previous studies ($v_p \sim 2 \mu\text{m}/\text{min}$, $v_d \sim 12 \mu\text{m}/\text{min}$, $f_c \sim 0.3/\text{min}$)? why would the depolymerization be slower in metaphase and the catastrophe frequencies almost 2 orders of magnitude higher? In fact, in other spindles there is evidence that the main change between interphase and metaphase is only an increase of the rescue rate at interphase (and catastrophes are around $\sim 1/\text{min}$). In any case, in metaphase MTs we would expect, if anything, equally or faster microtubule depolymerization velocity.

We agree with the Reviewer that the dynamic parameters were not well explained. We have now greatly expanded the discussion of this point in the revision. We began with measurements of MT dynamics in fission-yeast monopolar spindles that are now published (Blackwell et al, 2016 Biophys J), then modified them to increase the success of bipolar spindle assembly with Ase1 only. Our revised reference parameter set leads to a mean MT length of 0.6 micron, which is still quite short. However, when MTs are longer, problems with Ase1-only spindle assembly occur. We have discussed this point, and added further details in the text and Fig. 5, discussing how and why spindle assembly in this model is so sensitive to MT dynamics parameters.

-On the other hand, the parameters chosen lead to an unbound growth of Mts (Mt length diverging) for the stabilized Mts ($v_{sg} * s_{fr} * v_g * f_r - v_{svs} * s_{fc} * v_s * f_c > 0$, again from Verde, 92), which I am not sure it makes sense or if there is any evidence for that. Otherwise, could the authors provide any evidence for it? Equally important: How were the scaling factors fixed (their values seem very arbitrary)? As asked before, how would the results depend on these particular stabilization values?

As mentioned above, in the revised manuscript we have addressed this point by studying a greatly expanded range of parameters and examining their effects of spindle assembly and properties such as spindle length. In both our random parameter sets in which we sampled parameter values over broad ranges, and in our studies varying individual parameters with others fixed at the reference values, we found that stabilization of MT dynamics by crosslinking such that MT dynamics move into the unbounded growth regime is important for spindle assembly in the model. In particular, as shown in the revised figure 5 and in the supplementary figures, successful spindle assembly requires a significant increase in the rescue frequency and decrease in the shrinking speed upon crosslinking.

-The length scales for spindles obtained in the simulations are not consistent with the data. In the mutants, spindles are very small (far less than a micron from the figures), but in the simulation spindles reach a final length of $1.5 \mu\text{m}$, is this major discrepancy due to the chosen parameters? What is the limit size of the spindle in the simulation? does it stop growing because it is spatially constrained by the nuclear envelope?

We agree with the Reviewer that this discrepancy is important, and we have carefully studied this point in the revision. As mentioned above, we have studied the effects of the "wall force" in our model. Only when the wall force was zero were we able to reproduce the short spindles found experimentally. Then we were also able to demonstrate that spindle length increases as the number of Ase1 molecules is increased. In our model, spindle length is limited to a maximum of $2.75 \mu\text{m}$, because the SPBs are constrained to move on a sphere of this diameter.

-Dt and $D \backslash \theta$ of SPB come from unpublished data. Could the authors still give the gist of how Dt is obtained such that the reader can judge its validity?

The measurements of translational diffusion were performed by treating cells with MBC (Methyl benzimidazol-2-yl-carbamate) to depolymerize MTs in mitosis, then measuring the relative motion of the two fluorescently labeled SPBs to estimate their relative diffusion. The single SPB diffusion was then determined assuming that each SPB diffuses identically, and the rotational diffusion coefficient was estimated from the translational one based on the assumption that the surrounding fluid viscosities do not significantly affect the rotational diffusion. The paper containing these measurements is now *in press* (Blackwell et al, 2017 Science Advances).

Minor comments:

“force balance” mechanism is inappropriate. The system is always at force balance, but the force balance of each particular perturbation/situation may lead to a bipolar spindle or a monopole, for instance. Stating for example that kinesin-5 and kinesin-1 contribute to the balance of forces in the spindle is a more rigorous statement than that they are part of a force-balance system that can become unbalanced.

We agree with the Reviewer. This has now been corrected.

To be consistent with the notation throughout the supplement, vd,0 in Table 1 should be vs,0.

Corrected.

The manuscript will benefit from a figure where the schematics of the conceptual model are clearly shown.

Corrected.

Reviewer #4:

This manuscript presents an extensive study of the roles of various mitotic kinesin motors as well as the crosslinker Ase1 and the MT stabilizing protein Cls1 in spindle assembly and elongation in fission yeast. A combination of deletions, overexpressions etc. is used to map out the complex dynamics of spindle morphogenesis and activity in the later stages of mitosis. It was found earlier that deletion of the counteracting kinesin-5 and kinesin-14 motors can restore a functional, albeit sub-optimally performing spindle. The focus of this manuscript is on the mechanism of spindle assembly when those kinesins are absent. While the kinesin-6 motor Klp9 is shown to be essential for spindle elongation in anaphase, it is not this motor or any other motor that drives spindle assembly, but rather the combination of MT crosslinking by Ase1 and MT stabilization by Cls1. This is an interesting finding and demonstrates the redundancy and complex interaction between various cooperating and competing players in biomolecular machineries. How the spindle can be assembled just by bundling and stabilization is demonstrated in an also very complex numerical model using a combination of Brownian dynamics and Monte Carlo methods. Overall, this study is carefully done and presents very convincing and also surprising results. I recommend publication in Nature Communications after a few small issues are taken care of.

General issues:

- An important point made by the manuscript is that the numerical model proves that Ase1 and Cls1 are both necessary and sufficient for motor-less spindle assembly. I can accept that the experiments make a good case for the necessity. The modeling provides a quite suggestive mechanism to explain the experimental finding, but it is difficult to believe that the modeling proves the sufficiency. Given that the modeling relies on a large number of parameters and assumptions that are based on other experiments, which may or may not be

very accurate, one could imagine a large number of other players that are needed to make the system work.

- The essence of the model appears to me to be that antiparallel crosslinking and anchoring of the spb to the nuclear shell and anchoring of MT minus ends to the spb must result in an antiparallel spindle with opposing spbs (given the stiffness of MTs and MT bundles), unless the MTs fall apart so fast that the spindle has no time to form. It would be interesting to discuss in more detail than done now what the essential ingredients of the model really are to illuminate the general principle. There is a confusing multitude of parameters in the model many of which (such as the axial stiffness of Ase1 or the capture radius etc.) might not change the big picture very much when varied.

We agree with the Reviewers on both points. In the revision, we have studied how the model behavior varies with parameters in more detail, to identify the parameters that are truly important for spindle bipolarity. We have found that whether or not the model is able to form bipolar spindles is determined by the number of Ase1 molecules, the catastrophe frequency, the rescue frequency, the rescue frequency stabilization, and the shrinking speed stabilization. The extensive new text and Fig. 5 summarize our new findings.

- a further central result is that pushing forces by MT polymerization drive spindle assembly. It is not clear to me that these are really needed. When the spbs are already anchored to the nuclear shell they only need to slide to opposite sides. The crosslinking and bending stiffness of MTs (if they are long enough) should be sufficient to achieve that. Could one run the simulation without catastrophies and repolymerization, just with on-off of Ase1? There should definitely be an energy minimum with opposing poles and long antiparallel overlaps.

We performed the suggested simulations with the catastrophe frequency set to zero (with the wall force also set to zero, so that there was no force-induced catastrophe). We found that this version of the model was unable to assemble bipolar spindles, because the measured kinetics of Ase1 appear to be too slow to form MT antiparallel overlaps during initial growth. If MTs do not form antiparallel bundles when short, they do not show significant reorientation once they are long. It appears that the steric interactions between long MTs make their angular reorientation difficult. In addition, because Ase1 is able to bind to parallel MTs, when MTs become long parallel bundling takes over, due to the large number of Ase1 binding sites on nearby MTs emanating from the same SPB.

Minor points:

- title and abstract jump right into force balance in spindles without even mentioning mitosis. It might be good to start a bit more general with one sentence.

We have now made precise emphasis on mitosis both in the revised title, abstract, and introduction.

- Figure captions: "Each frame corresponds to 3 minutes interval." This is a bit obscure. I suppose it means these are all maximum intensity projections of 7 confocal planes taken with 300 ms exposure time at intervals of 3 min?

Yes, all our images are maximum projections of confocal z-stacks.

Supplementary Information:

There are numerous small typographic errors that should be corrected by careful proof reading, examples :

Pg 3: " only small dependence on the rotational diffusion coefficient to spindle structure."

Sentence needs correction

Pg 4: "We assume that, due to the relatively crosslinker concentrations", missing "low"?

Pg. 5: "Crosslinker that reach the end" should read: "Crosslinkers that reach the end"

We have reworked the supplement in the revision, and have carefully proofread it.

We thank again the four Reviewers for their time and effort in reading our revision. We hope that our paper is now acceptable for recommendation for publication.

Reviewer #1 (Remarks to the Author) .

The authors have responded to my original concerns...

1. in Figure 5 the word stabilization has a ?, and this needs to be fixed...

2. In figure 2D the red curves obscure the blue ones; perhaps change the colors? (for example 4C is easier to see all three)...

Reviewer #2 (Remarks to the Author) .

In the revised manuscript, the authors have addressed my concerns and questions. I think that this is an important study that significantly impacts our understanding of functional interactions between mitotic motors, microtubule crosslinkers and MT polymerases during the assembly and function of the mitotic spindle. It should be of interest to the journal readership and accordingly I recommend its publication in Nature Communications...

Reviewer #3 (Remarks to the Author) .

The authors have clarified my experimental concerns and some of the simulation concerns. Unfortunately I still have some concerns on the conclusions drawn from the simulations (see below). Overall, the message of the paper is important for the community and it is proven by the experiments, which are very sound. Therefore, I feel the paper should be published in Nature communications, but I think some issues with the simulations have to be clarified...

My major concern is the claim of necessity on MT stability and antiparallel bundling by ASE1 to form spindles in absence of motors (which is the header of a section of the paper). This claim is simply not justified with the work provided for several reasons. The foremost and most important is that the kinetochore-MT interactions are not considered. I think it is fine to leave that for future work as the authors argue, but since the possibility that this interaction may generate bipolar spindles in the absence of motors is not tested, then one cannot claim necessity. The justification in the text "Because fission yeast cells unable to form stable kinetochore-MT attachments can still assemble bipolar spindles 35, we have neglected any mechanical contributions of chromosomes." sounds reasonable when trying to check sufficiency, but not necessity, as in the assembly of these spindles the motors were present. The fact that the simulation is consistent with the experimental results is already very interesting and valuable, and I think the results as they stand cannot be pushed further than this...

Another issue I had during the previous revision was the unreasonably long length of the simulated spindles (and the choice of the dynamic parameters). This issue is not really resolved satisfactorily. For instance, the authors now write that in the presence of the wall force the spindles elongate to this unrealistic length, and only when the nuclear envelope force is set to 0 the spindles are short. To me that seems an inconsistency of the model because why wouldn't the microtubules not feel the envelope? At least a justification other than stating that without setting this force to 0 the right length cannot be obtained should be added. Arguing that MT-kinetochores may be the reason for the shortening the spindle argues for simulating the MT-kinetochores. The dynamic parameters for MTs lead to an unbound growth (as the authors have now added to the text), but since the force from the nuclear envelope is set to 0 now, does this mean the MTs have an unbound average length? Isn't the wall force necessary to trigger MT depolymerization? .

..The dynamics parameters have been modified from literature "to increase the success of bipolar spindle assembly with Ase1 only" (quoted from their response). I find that modifying the parameters to obtain the expected result is biasing the conclusions from the simulations. Adding to this, the justification to not use more reasonable parameters (that would lead to a reasonable MT length) because otherwise "problems with Ase1-only spindle assembly occur" is less than satisfactory: isn't the point of the simulation to show precisely that? perhaps the fact that

problems with spindle assembly occur using measured parameters points to more fundamental problems of the model..

..In summary, while the simulation is consistent with the assembly of bipolar spindles in the absence of motors, I find there are major issues in the simulations that are not consistent with the data. Therefore, I think it is necessary to tone down the conclusions drawn from the simulations and state only sufficiency or consistency..

Minor:..

..-Why is the criterium for a spindle to be viable to have an IF greater than 0.2 for longer than 2 minutes? And why IF is allowed to drop below 0.2 for 12 seconds? all these criteria sound a bit arbitrary and at least a rationale should be added..

- in the supplement there are many typos. For instance (and I stopped writing them down after these) in Table S1, asymptotic wall force and membrane tube radius seem to be lacking the proper reference. The second paragraph of the same page "Each the results of each". The last paragraph of section "parameter shotgun tests" on page 16 is incomplete..

Reviewer #4 (Remarks to the Author)..

The authors have carefully and extensively addressed the concerns of the referees, and the manuscript has clearly improved. I recommend publication in Nature Communications..

Rebuttal to the Reviewers

We thank the four Reviewers for their time and effort in reading our revised manuscript. In particular, we thank Reviewers 1, 2, and 4 for recommending publication.

Reviewer 3 had additional suggestions on the modeling aspect of the paper, which we have now addressed. Briefly, we have removed over-interpretation wordings, and have added explanatory text. Textual changes are highlight in yellow.

Below, we address specific points raised by the Reviewers.

Reviewer #1

The authors have responded to my original concerns.

1. In Figure 5 the word stabilization has a ?, and this needs to be fixed.
2. In figure 2D the red curves obscure the blue ones; perhaps change the colors? (for example 4C is easier to see all three).

We have made the suggested corrections. Thank you for recommending publication.

Reviewer #2:

In the revised manuscript, the authors have addressed my concerns and questions. I think that this is an important study that significantly impacts our understanding of functional interactions between mitotic motors, microtubule crosslinkers and MT polymerases during the assembly and function of the mitotic spindle. It should be of interest to the journal readership and accordingly I recommend its publication in Nature Communications.

Thank you for recommending publication.

Reviewer #3:

The authors have clarified my experimental concerns and some of the simulation concerns. Unfortunately I still have some concerns on the conclusions drawn from the simulations (see below). Overall, the message of the paper is important for the community and it is proven by the experiments, which are very sound. Therefore, I feel the paper should be published in Nature communications, but I think some issues with the simulations have to be clarified.

We have removed over-interpretation wordings, and have added explanatory text describing the model, as discussed below.

My major concern is the claim of necessity on MT stability and antiparallel bundling by ASE1 to form spindles in absence of motors (which is the header of a section of the paper). This claim is simply not justified with the work provided for several reasons. The foremost and most important is that the kinetochore-MT interactions are not considered. I think it is fine to leave that for future work as the authors argue, but since the possibility that this interaction may generate bipolar spindles in the absence of motors is not tested, then one cannot claim necessity. The justification in the text "Because fission yeast cells unable to form stable kinetochore-MT attachments can still assemble bipolar spindles 35, we have neglected any mechanical contributions of chromosomes." sounds reasonable when trying to check sufficiency, but not necessity, as in the assembly of these spindles the motors were present. The fact that the simulation is consistent with the experimental results

is already very interesting and valuable, and I think the results as they stand cannot be pushed further than this.

We agree and have removed the claim of necessity of Ase1. Ase1 is sufficient to form a bipolar spindle.

Another issue I had during the previous revision was the unreasonably long length of the simulated spindles (and the choice of the dynamic parameters). This issue is not really resolved satisfactorily. For instance, the authors now write that in the presence of the wall force the spindles elongate to this unrealistic length, and only when the nuclear envelope force is set to 0 the spindles are short. To me that seems an inconsistency of the model because why wouldn't the microtubules not feel the envelope? At least a justification other than stating that without setting this force to 0 the right length cannot be obtained should be added. Arguing that MT-kinetochores may be the reason for the shortening the spindle argues for simulating the MT-kinetochores. The dynamic parameters for MTs lead to an unbound growth (as the authors have now added to the text), but since the force from the nuclear envelope is set to 0 now, does this mean the MTs have an unbound average length?

Isn't the wall force necessary to trigger MT depolymerization?

The dynamics parameters have been modified from literature "to increase the success of bipolar spindle assembly with Ase1 only" (quoted from their response). I find that modifying the parameters to obtain the expected result is biasing the conclusions from the simulations. Adding to this, the justification to not use more reasonable parameters (that would lead to a reasonable MT length) because otherwise "problems with Ase1-only spindle assembly occur" is less than satisfactory: isn't the point of the simulation to show precisely that? Perhaps the fact that problems with spindle assembly occur using measured parameters points to more fundamental problems of the model.

In summary, while the simulation is consistent with the assembly of bipolar spindles in the absence of motors, I find there are major issues in the simulations that are not consistent with the data. Therefore, I think it is necessary to tone down the conclusions drawn from the simulations and state only sufficiency or consistency.

We will answer the questions raised above.

1. Regarding the wall force, as noted in the text, we believe that setting the wall force to zero helps the model overcome the limitations that arise from neglecting kinetochore attachments and chromosomes. In cells, MT-kinetochore interactions produce inward pulling forces that oppose the separation of sister kinetochores that tend to shorten the spindle. In the absence of these inward forces from chromosomes in our model, any wall force causes pushing forces when MTs grow into the wall that elongates the spindle. Therefore, we conceptualize setting the wall force to zero as compensating for the lack of chromosomes. We have added to this discussion in the text.
2. Regarding MT length, the length does not in fact become unbounded, because the stabilization that can lead to unbounded length only occurs when a crosslink is formed between two MTs with the crosslinker sufficiently close to the MT tip. When the MT grows away from the crosslinker, or the crosslink unbinds, the dynamics are destabilized and the MT shortens. Therefore, the actual MT length distribution is a weighted average of that corresponding to the bare MT dynamics parameters and the unbounded ones; the effect in practice is to lengthen the MTs but not cause the distribution to become unbounded.

3. The wall force is not necessary for MT depolymerization, because the MTs undergo catastrophe and shrink when a crosslink near the tip is lost.
4. Regarding alteration of parameters away from experimental values: we agree that this is a limitation of the model, necessitated, for example, by the absence of chromosomes and kinetochores in the model. We have added additional text to the discussion of the altered model parameters to emphasize this point. Further, we think that understanding how and why the MT dynamics parameters must be changed in our model to achieve bipolar spindle assembly remains of interest, even though the model does have simplifying assumptions.

Overall, we agree with the Reviewer. We have toned down by removing our previous claim of the necessity of Ase1 in bipolar spindle formation. Ase1 is now only sufficient but not necessary.

Minor:

-Why is the criterium for a spindle to be viable to have an IF greater than 0.2 for longer than 2 minutes? And why IF is allowed to drop below 0.2 for 12 seconds? all these criteria sound a bit arbitrary and at least a rationale should be added.

We agree that this was not explained well. We have added a rationale in the supplement. Briefly, we found by observing simulation results that if the IF remained high for at least 2 minutes, then the bipolar spindle was typically stable to the end of the simulation – typically unstable spindles would fall apart before the 2 minute period. Allowing drops of IF for up to 12 seconds was added because the IF is a fluctuating quantity which can transiently drop even for relatively stable spindles; adding this rule allowed the measurement to match spindles that we would consider a stable bipolar spindle by eye.

- In the supplement there are many typos. For instance (and I stopped writing them down after these) in Table S1, asymptotic wall force and membrane tube radius seem to be lacking the proper reference. The second paragraph of the same page “Each the results of each”. The last paragraph of section “parameter shotgun tests” on page 16 is incomplete.

We apologize for inadvertently submitting an earlier, uncorrected version of the supplement. We have proofread the supplement thoroughly and believe that the typos are now removed.

Reviewer #4:

The authors have carefully and extensively addressed the concerns of the referees, and the manuscript has clearly improved. I recommend publication in Nature Communications.

Thank you for recommending publication.

Reviewer #3 (Remarks to the Author) .

The authors have addressed and clarified all my concerns, and accordingly I recommend publishing the manuscript in Nature Communications..